# Inferring neuron-neuron communications from single-cell transcriptomics through NeuronChat

Wei Zhao [1], Kevin G. Johnston[1], Honglei Ren[1], Xiangmin Xu [2,3,4,5] & Qing Nie [1,3,5,6] ✉

Neural communication networks form the fundamental basis for brain function. These communication networks are enabled by emitted ligands such as neurotransmitters, which activate receptor complexes to facilitate communication. Thus, neural communication is fundamentally dependent on the transcriptome. Here we develop NeuronChat, a method and package for the inference, visualization and analysis of neural-specific communication networks among pre-defined cell groups using single-cell expression data. We incorporate a manually curated molecular interaction database of neural signaling for both human and mouse, and benchmark NeuronChat on several published datasets to validate its ability in predicting neural connectivity. Then, we apply NeuronChat to three different neural tissue datasets to illustrate its functionalities in identifying interneural communication networks, revealing conserved or context-specific interactions across different biological contexts, and predicting communication pattern changes in diseased brains with autism spectrum disorder. Finally, we demonstrate NeuronChat can utilize spatial transcriptomics data to infer and visualize neural-specific cell-cell communication.

Brain function relies on signal transmission among numerous neuronal and non-neuronal cells. The connectome—wiring organization of neural connectivity—is subject to transcriptional regulation[1,2]. Recent single-cell RNA-seq (scRNA-seq) datasets show heterogeneity for cell transcriptomic states[3], raising the possibility that differences in gene expression profiles within and across regions reflect neural signal processing states. The emerging methods of spatial transcriptomics[4,5], which measure the spatial locations of neural cells in addition to gene expressions in cells, also provide abundant resources for dissecting neuron heterogeneity. While most current analysis approaches for scRNA-seq and spatial data allow the classification of cell types, the capability to probe the intercellular communications which determine

the underlying anatomical and functional connectivity is still limited. Yet, these transcriptomics data inherently contain the expression of genes required for neural signal transmission, making it possible to infer such intercellular communications.

Recently, computational methods have been developed for inferring cell-cell communication networks from coordinated expressions of ligand-receptor interaction pairs[6–12] such as CellChat[9], CellPhoneDB[10] and NicheNet[11]. However, these methods are based on short-range autocrine/paracrine signaling which only acts through ligand diffusion or physical contact of cells[6]. Such approaches are not suitable for characterization of neuron-neuron communications because neurons can extend axons and dendrites over long distances

[1]Department of Mathematics and the NSF-Simons Center for Multiscale Cell Fate Research, University of California, Irvine, CA 92697, USA. [2]Department of Anatomy and Neurobiology, School of Medicine, University of California, Irvine, CA 92697, USA. [3]Department of Biomedical Engineering, University of California, Irvine, CA 92697, USA. [4]Department of Computer Science, University of California, Irvine, CA 92697, USA. [5]The Center for Neural Circuit Mapping, University of California, Irvine, CA 92697, USA. [6]Department of Developmental and Cell Biology, University of California, Irvine, CA 92697, USA. ✉e-mail: qnie@uci.edu

to form synapses and communicate mainly through neurotransmitter signaling[13–15]. Neurotransmitters, typically non-peptide small molecules such as glutamate and gamma-aminobutyric acid (GABA), are excluded from current protein-based ligand-receptor databases ligand–receptor databases[6–12,16]. For example, Smith et al. predicted 37 neuropeptide networks among cortical neuron types by taking the interaction score as the product of transcript levels of neuropeptide precursor and the cognate G-protein-coupled receptor[16–18], but did not include neurotransmitter signaling. Additionally, as small-molecule neurotransmitters are synthesized and transported into synaptic vesicles for fast release from the presynaptic neuron in response to stimulation, the abundance of small-molecule neurotransmitters used for synaptic transmission depends on the coordination of multiple genes such as synthesizing enzymes and vesicular transporters[19]. Overall, there is a lack of methods considering neurotransmitter signaling and system-level neural-specific cell-cell communications.

Here we present NeuronChat, a method that utilizes scRNA-seq data and/or spatially resolved transcriptomics to infer, visualize and analyze neural-specific cell-cell communication. Development of NeuronChat required manual curation of a new neural-specific database containing 373 entries of intercellular molecular interactions for both human (190) and mouse (183). By incorporating this database to model the coordinate expressions of cognate interacting molecules, NeuronChat infers the neural-specific communication networks among pre-defined cell groups from single-cell expression data. Through benchmark and applications to neural tissue datasets, we show NeuronChat's capability in revealing neuron-neuron interactions in several biological systems.

## Results

### Overview of NeuronChat

First, we curate a neural-specific database of intercellular molecular interactions for both mouse and human, named NeuronChatDB (Fig. 1a). Each interaction contains one ligand and a cognate target as well as the protein-coding genes related to their synthesis and vesicular transport. The ligands include small-molecule neurotransmitters, neuropeptides, gap junction proteins, gasotransmitters[20] and synaptic adhesion molecules, while the targets are typically but not limited to receptors. For example, the target proteins for neurotransmitters can also be uptake transporters or deactivating enzymes; the target proteins for gap junction proteins are other compatible gap junction proteins. For non-peptide neurotransmitters, corresponding synthesizing enzymes and/or vesicular transporters are included in the entry; for heteromeric receptors that contain multiple different subunits, corresponding subunits are curated into different entries with the same ligand. Among a total of 373 entries of ligand-target interaction pairs, there are 221, 73, 39, 16 and 24 entries related to small-molecule neurotransmitters, neuropeptides, gap junction proteins, gasotransmitters and synaptic adhesion molecules, respectively.

Second, we construct a computational model to link the expression data to cell-cell communication probability, based on coordinate expressions of interacting molecules of pre-defined cell groups (Fig. 1b). The input data for NeuronChat is a normalized cell-by-gene count matrix, with group annotations for cells. For each intercellular interaction pair in NeuronChatDB, we first average the expression level by cell group for all related genes, based on which we estimate the abundance of the ligand and the target for each cell group. For the non-peptide neurotransmitters, the genes contributing to ligand emission are first categorized into different biological function groups (e.g., synthesis and vesicular transport), and then the abundance is modeled by applying AND logic (i.e., geometric mean) among different groups of genes while applying OR logic (i.e., arithmetic mean) among redundant genes within the same group (see Methods); for other ligands and for all targets, the abundance is calculated as the average expression. Then, the cell-cell communication strength between two

groups is set to be the product of the ligand abundance of one cell group and the target abundance of another cell group. Significant communications can be determined by a permutation test where group labels of cells are randomly permuted and the communications strength is recalculated (see Methods). Thus, for each interaction, an intercellular communication network, i.e., a weighted directed graph composed of significant links between interacting cell groups, can be constructed. An aggregated communication network can be further obtained by summarizing all communication networks for individual interactions with four different aggregation methods (see Methods).

Third, we provide different methods for visualization and analysis of the inferred intercellular communication networks (Fig. 1c). Circle plot, chord diagram and heatmap can be used to visualize the communication strength among cell groups. NeuronChat can also perform quantitative analysis of the inferred communication networks to identify signaling patterns and categorize interactions. For multiple datasets from different biological contexts, NeuronChat can make systematic comparisons and identify conserved and context-specific ligand-target interaction pairs. For spatial transcriptomics data, NeuronChat can incorporate cellular spatial positioning into the inference of cell-cell communication and provides multi-layered visualization of spatial cell-cell communication.

### Benchmarking of NeuronChat

To investigate the ability of NeuronChat to predict intercellular communications, we first compare the predicted communication networks with those experimentally identified for benchmarking. Two cases were studied: (1) the projection network of the primary visual cortex (VISp) in mouse brain, and (2) the projection network of the anterior lateral motor cortex (ALM) in mouse brain. The connections from excitatory neurons of VISp and ALM to their cortical target regions were identified using monosynaptic retrograde labeling[3], where the viral tracers are injected into target regions and move towards the presynaptic neurons via retrograde axonal transport without further spreading to indirectly contacted cells, allowing the identification of direct neural connections[21–25]. By grouping retrogradely labeled neurons using their cell-type annotations, we obtain the coarse-grained projection networks composed of directed links from excitatory neuron types in VISp and ALM to their cortical target regions (Fig. 2a for VISp and Fig. 2f for ALM), which are then used for subsequent benchmarking. The single-cell RNA-seq data for VISp, ALM and their target regions are collected from two published papers[3,26]. The data used includes 6,785 glutamatergic cells of 7 subclasses (L2/3 IT, L4, L5 IT, L5 PT, L6 CT, L6 IT, and L6b) for VISp, and 13,824 glutamatergic cells from three cortical target regions (ACA, RSP and contralateral VISp); 3,883 glutamatergic cells of 5 subclasses (L2/3 IT, L5 IT, L5 PT, L6 CT, and L6 IT) for ALM, and 17,576 glutamatergic cells from the six cortical target regions (SSs, SSp, RSP, MOp, contralateral ALM and contralateral ORB).

By using NeuronChat, we infer the communication networks containing links from cell types of VISp (or ALM) to their target regions for all interaction pairs, and then aggregate them over all interaction pairs (see Methods for details). Next, we test whether this aggregated network predicts the projection network identified via retrograde labeling. We binarize the aggregate network by setting a threshold for the communication strength (Fig. 2b, g), to enable computation of the sensitivity (the fraction of links predicted by NeuronChat from those identified by retrograde labeling) and false-positive rate (the fraction of links predicted by NeuronChat from those NOT identified by retrograde labeling) for a given threshold. Repeating this process by scanning a set of continuous thresholds, we obtain the Receiver Operating Characteristic (ROC) curve. We then use the Area Under the ROC curve (AUROC) to measure NeuronChat's prediction performance: the closer to 1 the value, the better the prediction performance;

AUROC values for random classifiers are expected to be 0.5. The AUROC values are 0.832 and 0.764 for VISp (Fig. 2c) and ALM (Fig. 2h), respectively. Please note that a small portion of the communication links predicted are incorrect for both of the two cases (e.g., 3/21 for Fig. 2b and 7/30 for Fig. 2g). We also calculate another evaluation metric – the Area Under the Precision-Recall curve (AUPRC), which summarizes the trade-off between the recall (also known as sensitivity) and the precision (the fraction of links identified by retrograde labeling from those predicted by NeuronChat) under different thresholds. The projection networks predicted by NeuronChat show an AUPRC of

0.915 for VISp and an AUPRC of 0.768 for ALM, and are significantly better than the random classifiers (Fig. 2d, i). To determine whether the specific graph topology of ground truth labels (i.e., the projection network identified by retrograde labeling) makes the prediction task easy for NeuronChat, we perturb the ground truth labels by shuffling cell-type labels of graph nodes while keeping the same graph topology, and then recalculate AUROC and AUPRC. We find that the AUROC for the shuffled ground truth labels leads to a distribution with a mean of around 0.5, indicating a poor prediction ability for those shuffled labels even with the same topology (Supplementary Fig. 1a, c, e, g).

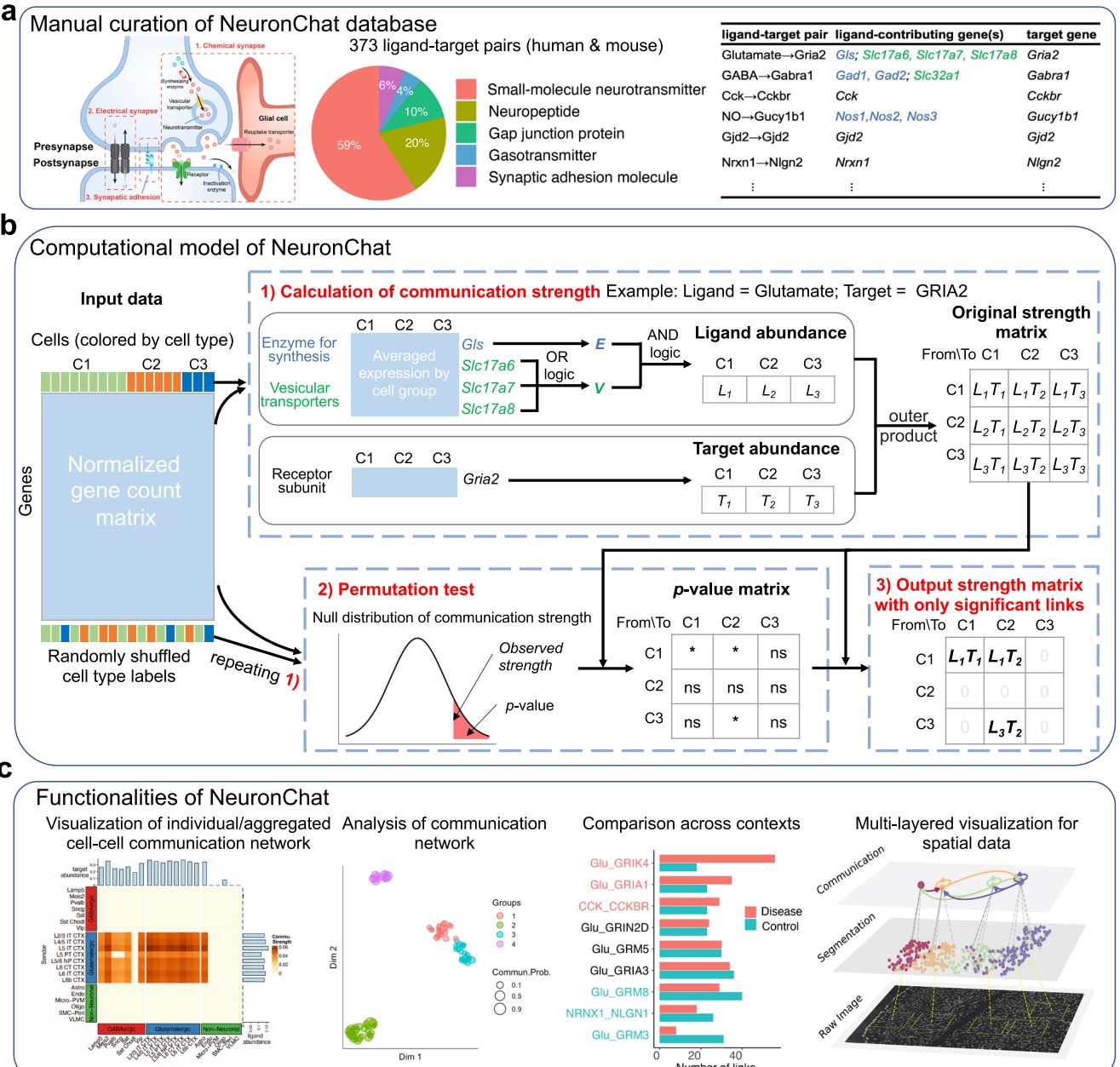

**Fig. 1 | Overview of NeuronChat. a** Overview of NeuronChat database. NeuronChat database includes ligand-target pairs required for chemical synapse, electrical synapse and synaptic adhesion (left panel). There are a total of 373 ligand-target pairs for both human and mouse, curated into five categories based on the type of the ligand (middle panel). The interaction pair list includes the ligand, target, and genes contributing to them (right panel). Note that genes contributing to the ligand are categorized into different groups (indicated by colors) based on their biological functions such as synthesis or vesicular transport. **b** Schematic diagram to illustrate the computational model of NeuronChat. The communication strength

characterizes the coordinated expression of genes required for ligand emission in the sender cell group, and the expression of the target gene in the receiver cell group. The statistical significance of a communication link is determined by the permutation test (* and ns represent significant and not significant, respectively). Only significant links are kept in the output communication strength matrix while values for not significant links are set to be zeros. See Methods for details. **c** Functionalities of NeuronChat: visualization and analysis of the intercellular communication networks, making systemic comparisons across different biological contexts, and multi-layered visualization for spatial transcriptomics.

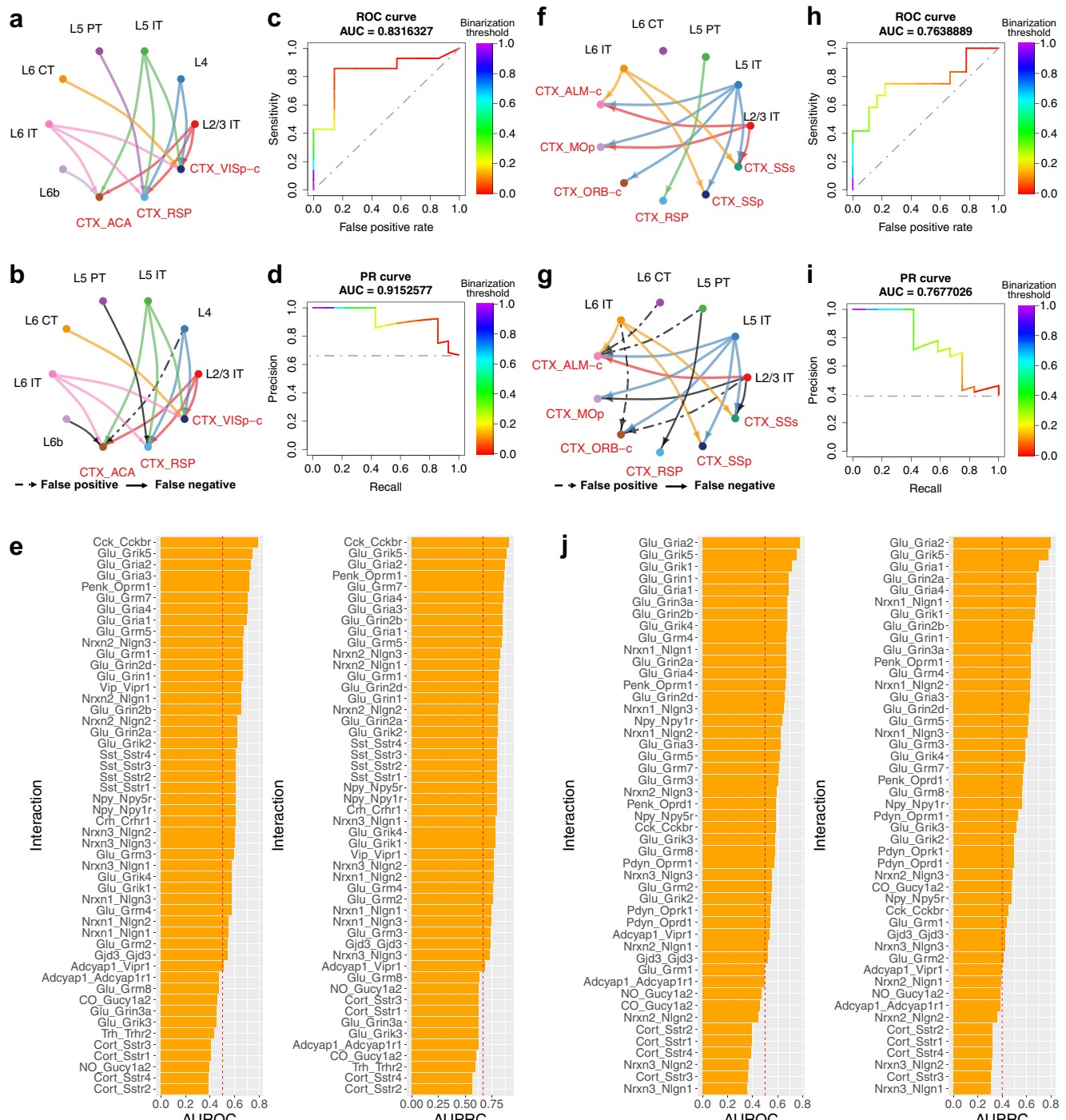

**Fig. 2 | Benchmarking NeuronChat on projection networks of two mouse cortex regions: VISp and ALM. a** Projections from seven cell types of VISp to their cortical target regions, identified by retrograde labeling[3]. **b** Aggregated and binarized intercellular communication network for VISp. The false-positive links represent those predicted by NeuronChat but not identified by retrograde labeling. The false-negative links are those identified by retrograde labeling but missed by NeuronChat prediction. Threshold for binarization is 0.028 (normalized by the maximum) in this plot. **c** ROC curve for the NeuonChat-inferred aggregated intercellular communication network for VISp, with AUROC indicated on the top.

The gray dashed line represents the random classifier with an AUROC of 0.5. The color bar indicates the binarization threshold. **d** PR curve for NeuonChat-inferred aggregated intercellular communication network for VISp, with AUPRC indicated on the top. The gray dashed line represents the random classifier with an AUPRC equal to the fraction of links identified by retrograde labeling (i.e., 14/21). **e** AUROC (left panel) and AUPRC (right panel) for the inferred VISp projection network of each individual interaction pair. **f–j** Repeat analysis for ALM, analogous to (**a–e**). Threshold for binarization is 0.165 for (**g**).

Furthermore, AUROC for the original ground truth labels is significantly higher than those for shuffled labels (*p*-values are 0.010 ± 0.0036 and 0.017 ± 0.0048 for VISp and ALM projection networks, respectively). Similar results are also obtained for the calculation of AUPRC (Supplementary Fig. 1b, d, f, h), suggesting

NeuronChat's prediction ability does not directly reflect the specific graph topology of ground truth labels.

In addition, the AUROC and AUPRC show little variations over input data subsampling rates ranging from 10% to 90% (Supplementary Fig. 2). While the predicted communication networks may

fluctuate among different repeated simulations due to finite sampling in the permutation test, the default number of permutations (100) used in NeuronChat yields consistent *p*-values and most of the significant links generated by 1000 permutations (Supplementary Fig. 3). Taken together, NeuonChat is relatively robust to the subsampling of input data and the number of permutations.

We also calculate the AUROC and AUPRC of the communication network for individual interaction pairs (Fig. 2e, j, for VISp and ALM, respectively). An interaction pair with a higher AUROC or AUPRC implies a better prediction of connectivity. As expected, the rankings of AUROC and AUPRC for individual interaction pairs are almost the same (for VISp: Spearman's rank correlation $\rho$ = 0.97, *p*-value <2.2×10$^{-16}$; for ALM: Spearman's rank correlation $\rho$ = 0.95, *p*-value <2.2×10$^{-16}$), indicating that an interaction pair with high AUROC usually shows high AUPRC. Out of top 10 interactions with highest AUROCs, 7 for VISp and 9 for ALM are mediated by glutamate, which is consistent with the fact that the projecting neurons are usually glutamatergic[27,28], and that glutamate is the major excitatory neurotransmitter in the brain. For example, the interaction between glutamate and *Grik5* (glutamate receptor, ionotropic kainate 5), and the interaction between glutamate and *Gria2* (one of the subunits of the AMPA receptor), are ranked top 3 in AUROC for predicting both VISp and ALM projections; the interactions between glutamate and other receptors such as *Gria1*, *Gria3*, *Grin1*, *Grin2b*, *Grm4*, *Grm7* and *Grik4*, also show high AUROC's for VISp or ALM. Additionally, the interaction between synaptic adhesion molecules Neurexin 1 (*Nrxn1*) and Neuroligin 1 (*Nlgn1*), which connect pre- and postsynaptic neurons respectively and play a vital role in synapse formation and maturation[29], is ranked #10 in AUROC and #6 in AUPRC for predicting ALM projections. Interestingly, Nrxn2-Nlgn2 interaction shows moderate prediction ability for VISp projections although neuroligin 2 is believed to locate on the inhibitory synapses[30]. Surprisingly, the interaction between neuropeptide *Cck* (cholecystokinin) and its cognate receptor *Cckbr* shows the highest prediction ability for VISp projection, indicating that the Cck-Cckbr interaction could be related to long-range neuronal connectivity. The interaction between neuropeptide *Penk* and its cognate receptor *Oprm1* also shows high AUROC for both VISp and ALM. Thus, when experimental connectivity data is available, NeuonChat is able to uncover biologically meaningful interaction pairs that are related to neural connectivity; when experimental connectivity data is not available, the interaction pairs with high information flow (defined as the sum of communication strength over all detected significant links), can be potential candidates underlying neural connectivity, as AUROC (or AUPRC) is found to have a moderate positive correlation with information flow (Supplementary Fig. 4).

Next we use examples to illustrate how NeuronChat utilizes variations in the abundance profiles of ligands in sending cell groups and/or targets in receiving cell groups to differentiate the communication strength. While all types of glutamatergic neurons use glutamate as the major neurotransmitter, the ligand abundance profiles show clear differences among the sending cell groups in VISp (Supplementary Fig. 5a, upper panel): L2/3 IT, L4, L5 IT, and L6 IT are more abundant in most of the ligands than L5 PT, L6 CT and L6b. Likewise, the target abundance profiles show large diversity in the receiving cells of different target regions (Supplementary Fig. 5a, lower panel): the major targets are most abundant in contralateral VISp while least abundant in ACA. These overall differences are consistent with the fact that L2/3 IT, L4, L5 IT and L6 IT have more outgoing links than other sending cell groups, and contralateral VISp and RSP have more incoming links than ACA (Fig. 2a, b). At the individual interaction level, for example, for the Glu-Gria2 interaction pair, the relatively high expression of genes related to glutamate synthesis and transportation in L2/3 IT, L4, L5 IT and L6 IT makes these cell types the major senders of the inferred communication network; contralateral VISp and RSP express higher *Gria2* than ACA, and are thus inferred as the major receivers

(Supplementary Fig. 5b). This pattern is even clearer for Cck-Cckbr interaction pair (Supplementary Fig. 5c). The analysis for ALM projection networks is also carried out, showing similar results (Supplementary Fig. 6). Taken together, variations in the abundance profiles of ligands in sending cell groups and/or targets in receiving cell groups allow NeuronChat to differentiate the communication strength.

## Comparison with other cell–cell communication inference tools and modeling settings

We further compare NeuronChat with two popular cell–cell communication inference (CCI) tools CellChat and CellPhoneDB in predicting neuronal connectivity using the same ligand-target database (i.e., NeuronChatDB). We use the same computational workflow of NeuronChat for the implementation of CellChat and CellPhoneDB except for the calculation of ligand abundance and the formula for communication strength (see details in Methods for comparison). On the inference of neuronal connectivity in both VISp and ALM projection networks (see the section above), NeuronChat outperforms existing CCI methods in two ways: (1) for the aggregated communication network, NeuronChat has the highest AUROC and AUPRC among three benchmarking methods (Fig. 3a, b for VISp and Fig. 3f, g for ALM); (2) for individual communication networks, NeuronChat not only detects more interaction pairs but also yields higher AUROC and AUPRC than the other two methods (Fig. 3c–e for VISp and Fig. 3h–j for ALM). Because both CellChat and CellPhoneDB use AND logic (geometric mean or minimum) rather than OR logic for redundant genes for the same function, the abundance of the small-molecule neurotransmitter is dramatically underestimated, leading to fewer detected interaction pairs than NeuronChat. These results demonstrate the advantage of NeuronChat in predicting neural-specific cell-cell communications.

While NeuronChat uses expressions of synthesizing enzymes and vesicular transporters to estimate the abundance of small molecular neurotransmitters, we note that a recent method scFEA[31] uses a graph neural network model to estimate metabolic flux and balance from scRNA-seq by incorporating stoichiometric effects of metabolism and pathways dependency. To investigate the effects of different abundance surrogates for small-molecule neurotransmitters in identifying neural-specific communication networks, using glutamate as an example, we compare NeuronChat's ligand abundance and eight scFEA-derived surrogates (including metabolite balance and seven module fluxes) in predicting VISp and ALM projection networks. For each of the nine glutamate surrogates, we calculated AUROC and AUPRC values for the communication networks of 24 glutamate-mediated interaction pairs, and found that NeuronChat's ligand abundance shows middle or above ranking in AUROC (or AUPRC) median among the nine glutamate surrogates (Supplementary Fig. 7a, c). For the communication network aggregated over 24 glutamate-mediated interaction pairs, NeuronChat's ligand abundance ranks #2 in both AUROC and AUPRC for predicting VISp and ALM projecting networks among the nine glutamate surrogates (Supplementary Fig. 7b, d). Nevertheless, the difference between NeuronChat's ligand abundance and the best scFEA-derived surrogate is very minimal. These results indicate that NeuronChat's ligand abundance works well on the inference of neuronal connectivity despite its simplicity.

NeuronChat uses Tukey's trimean (see Methods) rather than arithmetic mean to calculate the average of the gene expression in all cells of a group. According to their definitions, for a given gene and a given cell group, the non-zero Tukey's trimean only occurs if the gene is expressed in at least 25% of cells while non-zero arithmetic mean occurs if the gene is expressed in at least one cell. Because the genes only expressed in a small proportion (less than 25%) of cells are filtered out, Tukey's trimean benefits to identify the cell-type enriched ligand–target pairs. As expected, Tukey's trimean leads to fewer detected interaction pairs than arithmetic mean (Supplementary Fig. 8a, e); however, the interaction pairs produced by Tukey's

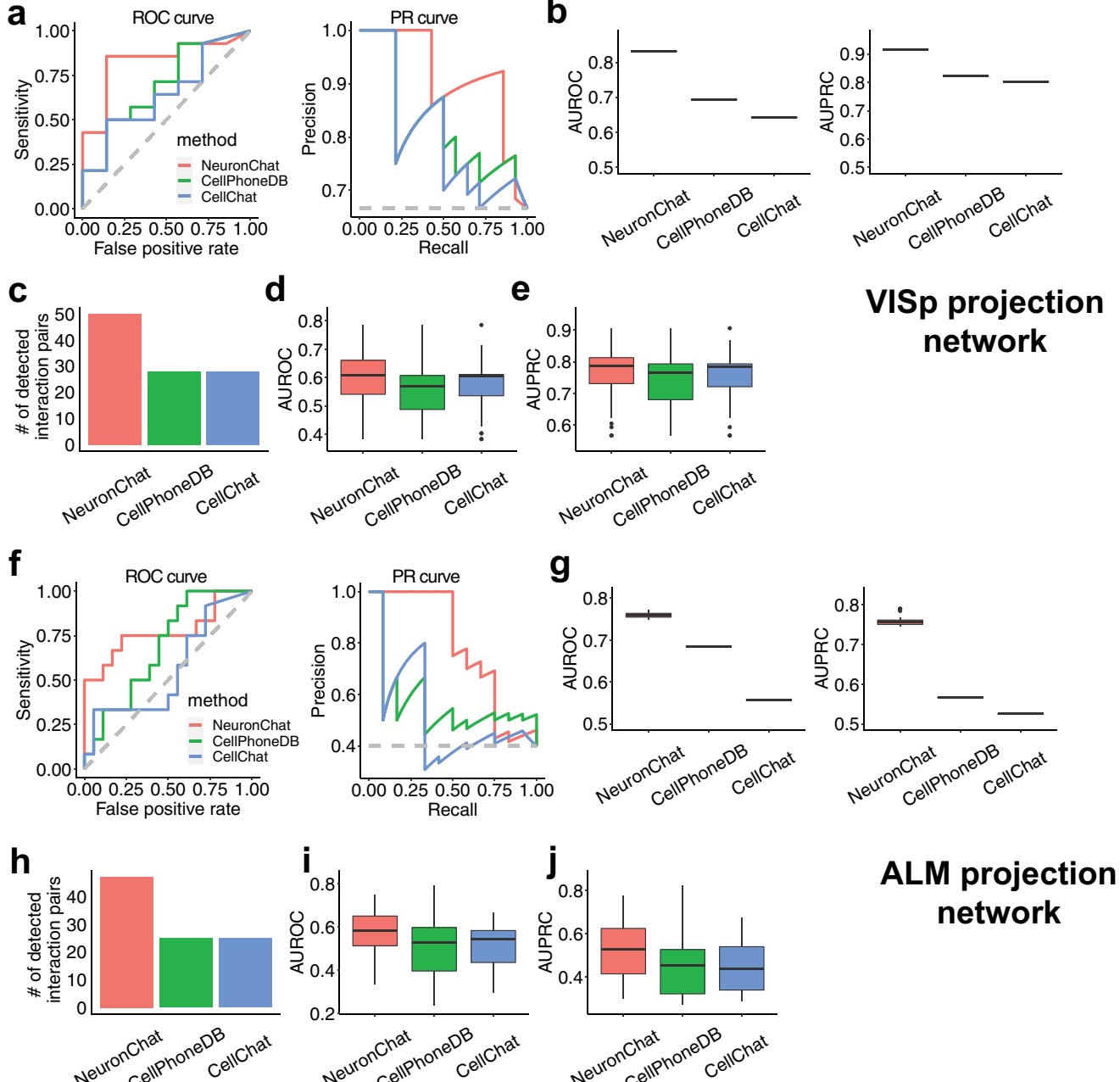

**VISp projection network**

**ALM projection network**

**Fig. 3 | Comparison of NeuronChat, CellPhoneDB, and CellChat in predicting VISp and ALM projection networks. a** Typical ROC curves (left panel) and PR curves for the three methods. **b** The boxplots of AUROC (left panel) and AUPRC (right panel) values for 100 repeats of the aggregated VISp projection networks inferred by the three methods. Each boxplot represents 100 independent repeated computations. Boxplot elements: center line, median; box limits, upper and lower quartiles; whiskers, 1.5x interquartile range; points, outliers. Note that no variation in each boxplot is observed because the aggregated method "thresholded_weight" reduces the fluctuation caused by finite sampling in the permutation test (see also Supplementary Fig. 9). **c** The number of detected interaction pairs for the three methods. **d, e** The boxplots of AUROC (**d**) and AUPRC (**e**) values for the individual VISp projection networks inferred by the three methods. Sample size (i.e., the number of detected interaction pairs) for boxplot: $n$ = 50, 28, and 28 for NeuronChat, CellPhoneDB and CellChat, respectively. Boxplot elements: center line, median; box limits, upper and lower quartiles; whiskers, 1.5x interquartile range; points, outliers. **f–j** Repeat analysis for ALM projection network, analogous to (**a–e**). Sample size for boxplot: $n$ = 47, 25, and 25 for NeuronChat, CellPhoneDB and CellChat, respectively.

trimean show overall higher AUROC and AUPRC than those produced by arithmetic mean, suggesting Tukey's trimean is able to infer more reliable interaction pairs (Supplementary Fig. 8b–d and 8f–h).

We also investigate the effects of four different aggregation methods on inferring neuronal connectivity. For the "thresholded weight" method, we choose the threshold for an interaction pair as the 80% quantile of all communication strength values for the interaction pair (the default setting for benchmarking). This is because the 80% quantile leads to overall higher AUROC/AUPRC than other thresholding quantiles, except a slightly lower AUROC for ALM projection network (Supplementary Fig. 9a, b and 9e, f). Among the four aggregation methods, we find that the "thresholded weight" method produces the highest AUROC/AUPRC values for the VISp projection network, and the second highest AUROC/AUPRC values that are only slightly lower than the best ones for the ALM projection network (Supplementary

Fig. 9c, d and 9g, h). Furthermore, "thresholded weight" leads to smaller variations in AUROC/AUPRC than other aggregation methods for repeated simulations, thus robustly minimizing the randomness generated in the permutation test.

## NeuronChat identifies intercellular communication patterns and functional-related interactions in VISp

The basic function of NeuronChat is to infer and visualize the intercellular communication networks, and then identify intercellular communication patterns and categorize the functionally related interaction pairs. We illustrate these functionalities of NeuronChat by applying it to the single-cell RNA-seq data of mouse VISp[3]. The 15,469 single cells are well annotated by three cell classes (Glutamatergic, GABAergic, and Non-Neuronal) and 21 cell subclasses. By applying NeuronChat to this dataset, we identify the communication networks among the 21 cell subclasses, which can be visualized by the circle plot, chord diagram and heatmap (Fig. 4a). Each of the glutamatergic subclasses sends signals to most of the GABAergic subclasses and all glutamatergic subclasses as well as astrocytes, while the communication strength for these links may differ. Interestingly, there are dense communications among glutamatergic subclasses. Compared to glutamatergic subclasses, GABAergic subclasses show relatively sparse outgoing communications to the three cell classes. Among GABAergic subclasses, *Lamp5*, *Sncg*, and *Vip* are the major senders; while *Lamp5*, *Pvalb* and *Vip* subclasses receive signals from both GABAergic subclasses and glutamatergic subclasses, *Sst* and *Meis2* subclasses show a preference for receiving signals from glutamatergic neurons.

A total of 109 significant interaction pairs are detected along with the number of links for all interaction pairs (Supplementary Fig. 10). After computing the information flow for each interaction pair, the interaction pairs Nrxn1-Nlgn1, Nrxn3-Nlgn1, Glu-Gria2, Glu-Grin2b, GABA-Gabra1, Glu-Gria4, GABA-Gabrb1, Glu-Grm5, GABA-Gabrg2 and Glu-Grik2 are ranked top 10 in the information flow, while NO-Gucy1a2, Glu-Gria2, Glu-Grin1, Glu-Grin2b, GABA-Gabbr1, GABA-Gabrb1, Nrxn1-Nlgn1, GABA-Gabrb3, Glu-Gria1 and Nrxn3-Nlgn1 are ranked top 10 in the number of links. The incomplete overlapping between the top 10 in the information flow and the top 10 in the number of links, indicates that some interactions (e.g., Glu-Gria4) are specific with strong individual links, while some others (e.g., Glu-Grin1) show wide communications among cell subclasses but moderate strength of individual links.

Using a pattern recognition method[32,33] (see Methods), NeuronChat detects the outgoing patterns of sending cells and the incoming patterns of receiving cells, which can be visualized via alluvial plots. This enables visualization of the correspondence between sending/receiving cell types and latent patterns, and the correspondence between latent patterns and individual interaction pairs (Fig. 4b). As expected, all glutamatergic subclasses correspond to the outgoing pattern #1, which is related to all glutamate signaling, *Nrxn1* signaling, and a few of neuropeptide interactions (e.g., Cck-Cckbr and Adcyap1-Adcyap1r1). In line with the diversity of inhibitory neurons[34], GABAergic subclasses correspond to outgoing patterns #2-3: *Lamp5, Sst* and *Sst Chodl* subclasses belong to pattern #2, which is mainly related to the signaling of neuropeptides *Sst, Npy* and *Cort*; *Pvalb, Meis2, Sncg, and Vip* subclasses belong to pattern #3, which includes signals of GABA, glycine and *Nrxn3* as well as some neuropeptides such as *Vip* and *Pnoc*. For non-neuronal subclasses, Endo, Micro-PVM and SMC-Peri correspond to outgoing pattern #2, while Astro and VLMC belong to pattern #4 that represents interactions of gap junction proteins. Different from the outgoing patterns, the incoming patterns #2 and #4 include both glutamate signals and GABA signals, indicating that corresponding cell subclasses (e.g., L5 IT CTX and Lamp5) receive both excitatory and inhibitory inputs. Nevertheless, glutamatergic subclasses and GABAergic subclasses

correspond to different incoming patterns (patterns #1 & #4, and patterns #2-3, respectively).

Through manifold learning[35] (see Methods), NeuronChat projects the interaction pairs into a two-dimensional manifold and groups them into different clusters based on the functional similarity of the communication networks. Functional similarity measures the degree to which interaction pairs share similar senders and receivers. The 109 interaction pairs are classified into 5 separate groups (Fig. 4c, d). Group #1, dominated by glutamate signals, represents the signaling from glutamatergic subclasses, while Group #4 includes many GABA and glycine signals and represents the signaling from GABAergic subclasses (see also the aggregated communication network for each interaction group from Supplementary Fig. 11). Interactions between neurexins and neuroligins dominate Group #3. Group #5 contains signaling of neuropeptides such as *Sst, Vip, Penk* and *Npy* from GABAergic subclasses, indicating that these interaction pairs share similar communication patterns. Group #2 is dominated by gap junction proteins and some neuropeptide signals, largely representing the signaling among non-neuronal subclasses. The results of manifold learning demonstrate that NeuronChat is able to categorize the detected ligand-target interaction pairs into biologically meaningful groups.

## NeuronChat reveals conserved and context-specific communication patterns between interlaminar excitatory networks for ALM and VISp

Another application of NeuronChat is to make comparisons across different biological contexts, to identify communication patterns conserved or specific to contexts. Here, we illustrate such functionality of NeuronChat by comparing the interlaminar excitatory communication networks between ALM and VISp. The single-cell RNA-seq data are from the published paper[26], and include 4,600 glutamatergic neurons for ALM and 8,114 for VISp, both of which are grouped into 8 shared subclasses. While the overall communication patterns are similar for both (Fig. 5a), some communication networks for individual interaction pairs are different. For example, Glu-Grin3a communication networks for ALM and VISp show a dramatic difference (Fig. 5b): L4/5 IT CTX is predicted to be one of the major target cell subclasses for ALM but not for VISp.

To identify the difference in individual communication networks, we compare the number of links and information flow for all individual interactions between the two regions (Fig. 5c). We find that only a few interaction pairs share the equal or near-equal information flow (or number of links) between ALM and VISp, e.g., Glu-Grin1 and Glu-Grm5. Other interaction pairs show significant differences between the two regions, for example, *Pdyn* signaling (including Pdyn-Oprm1, Pdyn-Oprk1, and Pdyn-Oprd1) is exclusively ON in ALM; Glu-Grik4 and Glu-Gria1 communication networks are much denser in ALM than in VISp; Glu-Grm3 and Glu-Grm7 communication networks as well as those mediated by *Nlgn1* show more links and higher information flow in VISp than in ALM.

Another way to compare the individual communication networks for ALM and VISp is based on their functional similarity. By projecting the interaction pairs of VISp and ALM onto the same two-dimensional manifold according to their functional similarity and then categorizing them into clusters, we can further spot the interaction pairs that are conserved or context-specific (Fig. 5d). If the communication patterns for one interaction pair are conserved between ALM and VISp, then the communication networks for such interaction pairs for ALM and VISp should be grouped into the same cluster, and vice versa. The two-dimensional manifold shows that all of the ligand-target interaction pairs are categorized into five clusters, and the aggregated network for each cluster is shown by heatmaps (Fig. 5e): cluster #1, related to neuropeptide signaling (e.g., *Sst, Trh, Nmb, Pdyn* and *Vip*), represents a sparse communication pattern dominated by the signal from L6 IT CTX

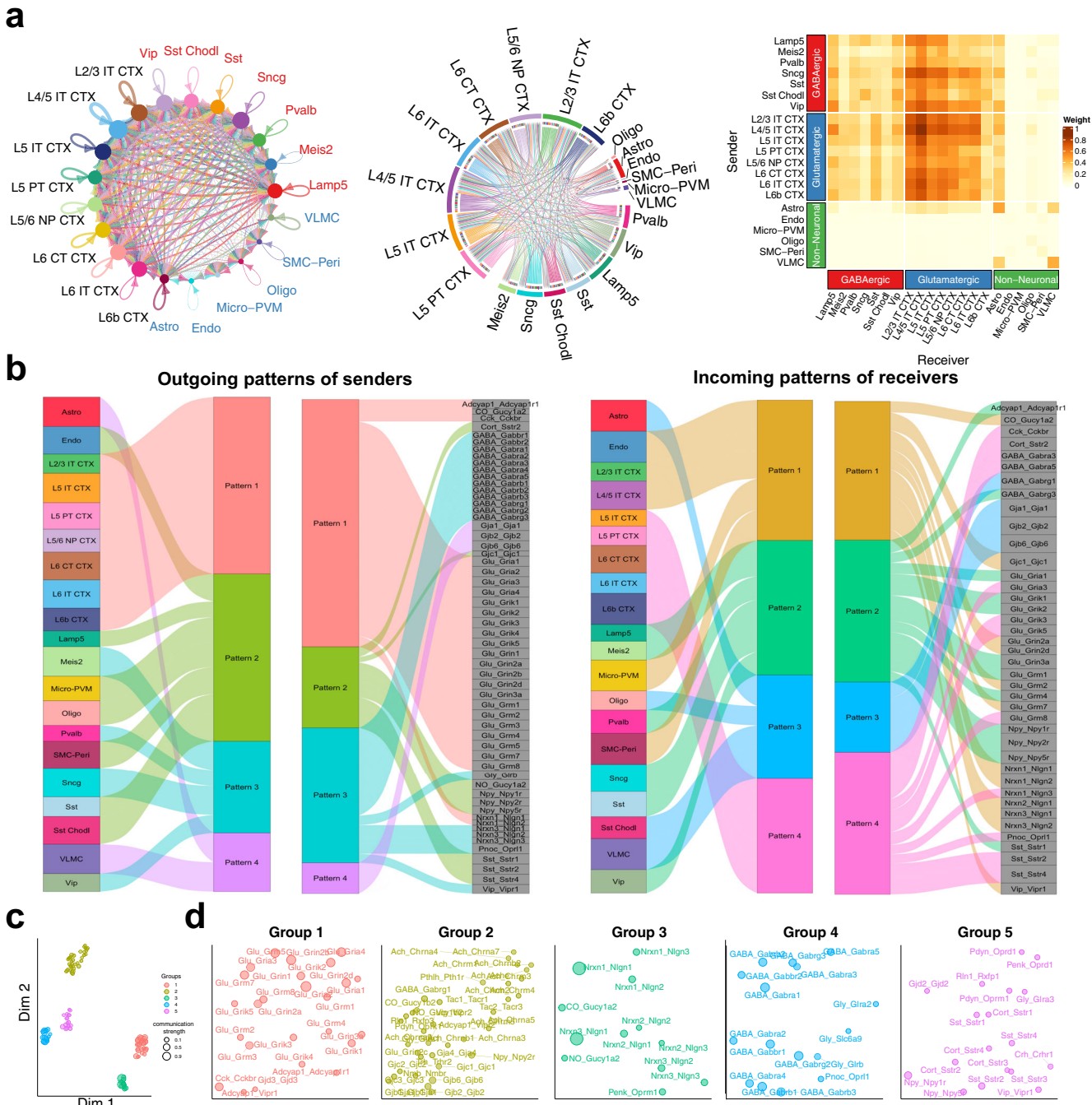

**Fig. 4 | Visualization and analysis of intercellular communication networks for mouse primary visual cortex. a** Visualizations of the inferred communication networks among multiple cell types of VISp. Circle plot, chord diagram and heatmap are used to visualize the intercellular communication networks aggregated by method "weight". In the circle plot, the node size indicates the strength of the outgoing signal from the cell, and the text labels are labeled by different colors to indicate their cell classes (i.e., glutamatergic, GABAergic and non-neuronal); the communication strength for each link is indicated by the width. In the chord diagram, sectors for different cell classes are repelled by larger gaps; the width of each link represents the communication strength, while the width of a sector (representing a cell group) reflects the strength of the total communications from or to this cell group. The color bar in the heatmap indicates the communication strength.

**b** The outgoing signaling patterns of senders and incoming signaling patterns of receivers visualized by alluvial plots, which show the correspondence between the inferred latent patterns and cell groups, and the correspondence between the inferred latent patterns and interaction pairs. The thickness of the flow indicates the contribution of the cell group or interaction pair to each latent pattern. The height of each pattern is proportional to the number of its associated cell groups or interaction pairs. The top 60 interaction pairs with the highest information flow are used for analysis. **c** Projecting interactions onto a two-dimensional manifold according to their functional similarity. Each interaction pair is represented by a dot, whose color and size indicate the functional group and the total communication strength (normalized by the maximum), respectively. **d** Magnified view of the two-dimensional manifold for each interaction group.

to L6b CTX; cluster #2, including glutamate signaling and neurexin-neuroligin interactions, shows the communication pattern where L2/3 IT CTX, L4/5 IT CTX and L5 PT CTX are the major receivers; cluster #3 represents the communication pattern, in which the signal mainly

comes from L2/3 IT CTX, L4/5 IT CTX and L5 PT CTX; cluster #4 shows a clear pattern where the L4/5 IT CTX and L5/6 NP CTX receive the signals from all subclasses; cluster #5 represents dense and strong communications among subclasses, which is dominated by glutamate signals.

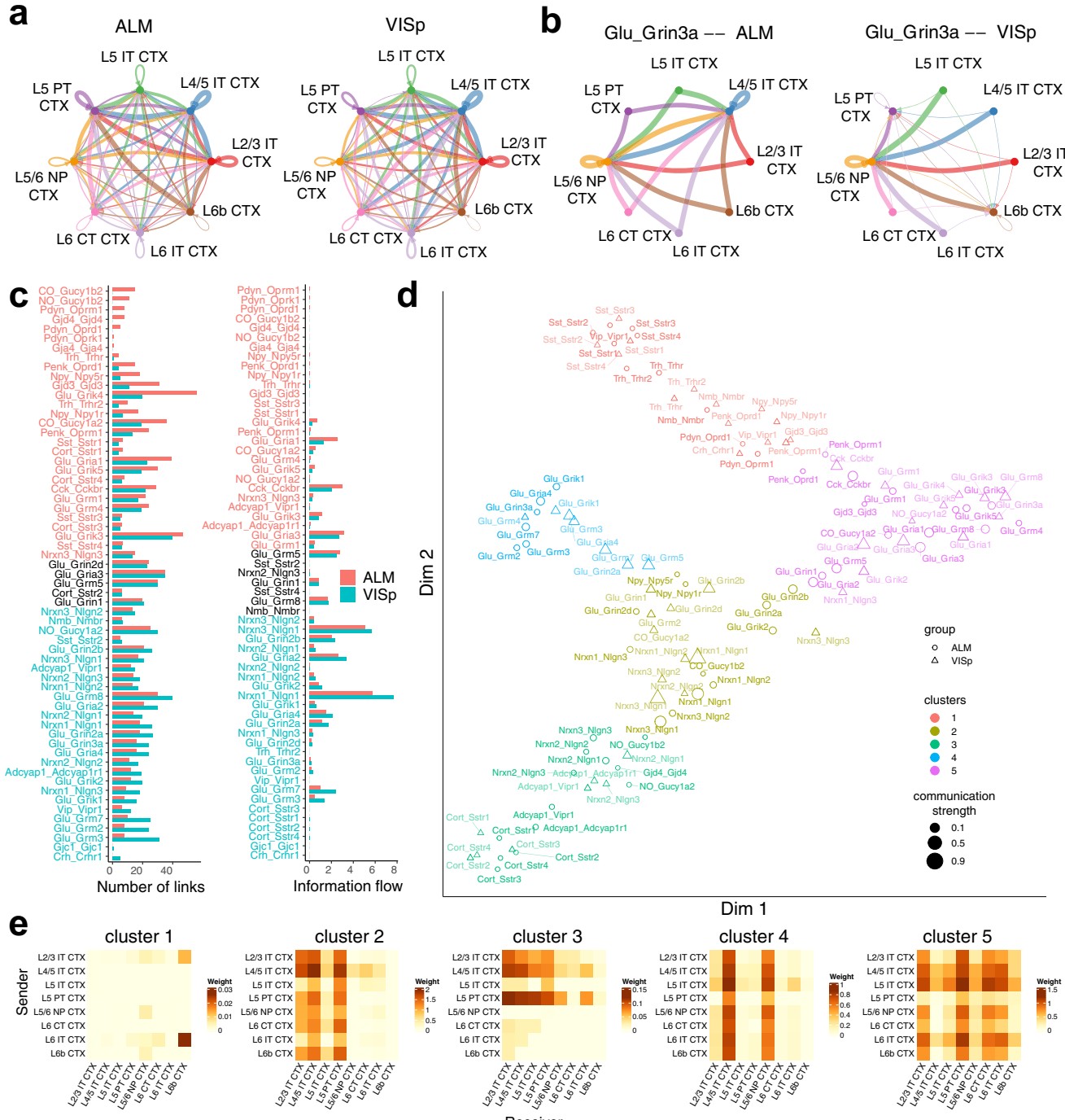

**Fig. 5 | Comparison analysis of interlaminar excitatory communication networks for ALM and VISp. a** The aggregated interlaminar excitatory communication networks for ALM (left panel) and VISp (right panel). The aggregation method used is "weight". **b** The individual interlaminar excitatory communication networks of Glu-Grin3a for ALM (left panel) and VISp (right panel). **c** Bar charts comparing the number of links (left panel) and information flow (right panel) between ALM and VISp for each interaction pair. **d** Projecting interactions for ALM and VISp onto a two-dimensional manifold according to their functional similarity. Each interaction pair is represented by a circle (for VISp) or triangle (for ALM), whose color and size indicate the functional group and total communication strength (normalized by the maximum), respectively. **e** Heatmaps of the aggregated interlaminar excitatory communication networks for the five interaction clusters in (**d**).

Interestingly, for most of the ligand-target interaction pairs, the communication networks for ALM and VISp are categorized into the same clusters, indicating most ligand-target interaction pairs may be conserved between the two regions. For example, communication networks for Glu-Gria2 in the ALM and VISp are both grouped into cluster #5.

However, the communication networks of Glu-Grin3a for ALM and VISp are grouped into cluster #4 and cluster #5, respectively, indicating that Glu-Grin3a is context-specific for ALM and VISp (see also Fig. 5b); likewise, the communication networks of Nrxn1-Nlgn3 for ALM and VISp belong to cluster #2 and cluster #5, respectively.

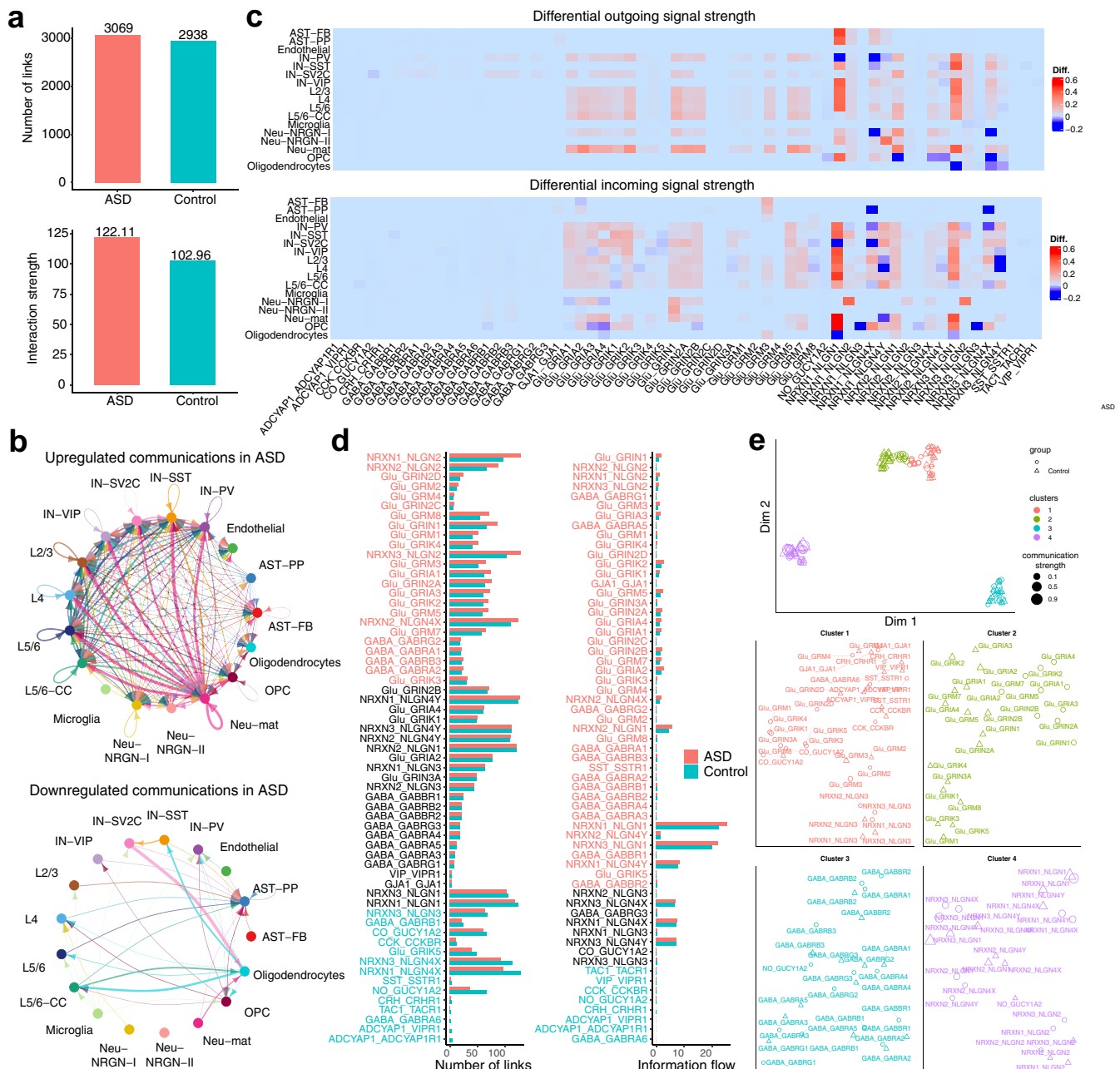

**Fig. 6 | Comparison analysis of cell-cell communications in cortex between ASD patients and controls. a** Comparison of the number (left panel) and weight (right panel) of all the links inferred between ASD and control. **b** Circle plots showing the upregulated (upper panel) and downregulated (lower panel) intercellular communications in ASD compared to control. The width of each link indicates the absolute difference between ASD and control, in the sum of communication strength values over all interaction pairs. **c** Heatmap showing the differential outgoing (upper panel) and incoming (lower panel) signal strength between ASD and control. Given the cell group and interaction pair, the outgoing (or incoming) signal strength is defined as the sum of communication strength over the links from (or to) the cell group. The color bar indicates the difference in the outgoing (or incoming) signal strength values between ASD and control. **d** Bar charts comparing the number of links (left) and information flow (right) between ASD and control for each interaction pair. **e** Projecting interactions onto a two-dimensional manifold according to their functional similarity (upper panel) and magnified view of each interaction group (lower panel). Each interaction pair is represented by a circle (for ASD) or triangle (for control), whose color and size indicate the functional group and total communication strength (normalized by the maximum), respectively.

## NeuronChat predicts the change of intercellular communication patterns in patients with autism spectrum disorder

NeuronChatDB also includes the ligand-target interaction information of humans. We investigate how NeuronChat can be used to predict the change of intercellular communication patterns in patients with particular neurological diseases, by applying it to the published single-nucleus RNA sequencing data of cortical tissue from patients with autism spectrum disorder (ASD) and healthy controls[36]. The data were collected from postmortem tissue samples including prefrontal cortex and anterior cingulate cortex from 15 ASD patients and 16 controls, containing 52,003 single nuclei for ASD patients and 52,556 for controls. To reduce the computational cost, a total of 20,000 cells are sampled for analysis (10,000 for ASD and 10,000 for control).

We use NeuronChat to infer the communication networks among 17 cell types, including subtypes of excitatory neurons, interneurons and astrocytes, for both ASD patients and controls. For the

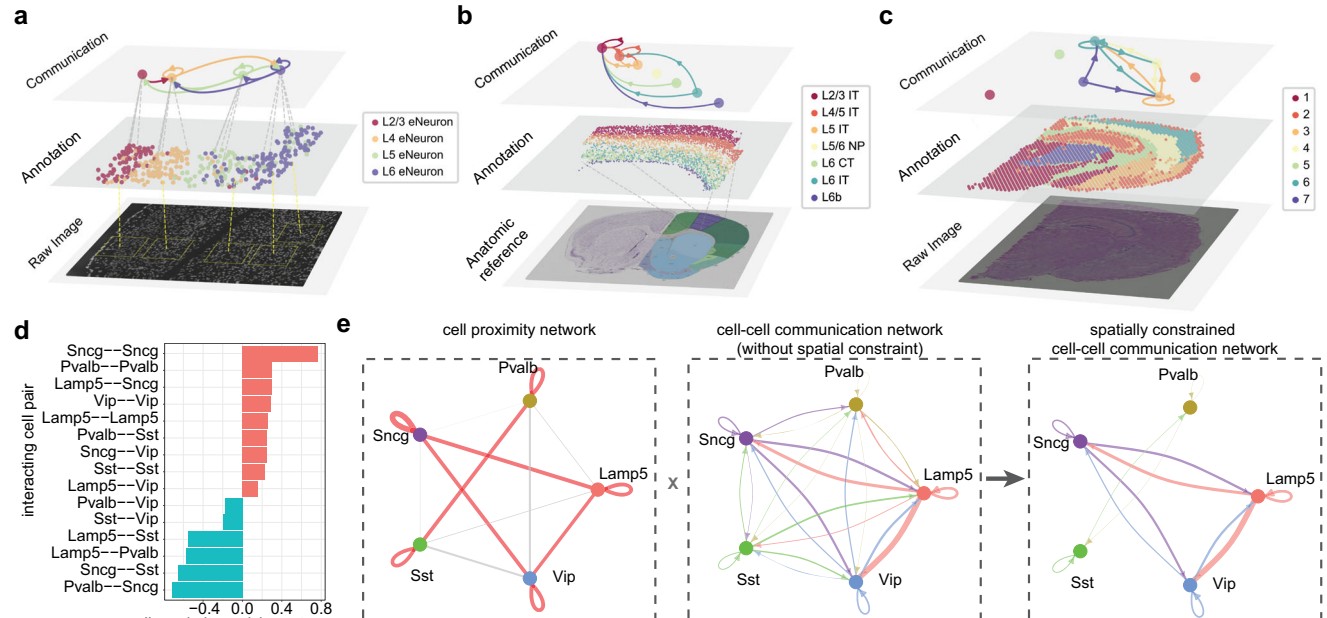

**Fig. 7 | Multi-layered visualization for spatial data and inference of spatially constrained communication network. a–c** Multi-layered visualization for three spatial transcriptomics datasets generated by different techniques seqFISH+ (**a**), MERFISH (**b**), and Visium (**c**). Each plot includes the raw tissue slice image/anatomic reference (bottom), cell/spot annotation in space (middle), and the aggregated communication network with the top 10 links shown (top). The width of a link indicates the sum of communication strengths over all significant ligand-target pairs. See Supplementary Fig. 12 for the full aggregated networks. The bottom image in (**b**) is the brain anatomic reference (Allen Mouse Brain Atlas, mouse.brain-map.org[56] and Allen Reference Atlas atlas.brain-map.org[57]). **d** Bar plot showing the cell proximity enrichment scores for all pairwise interacting cell types. The cell proximity enrichment scores are calculated based on all 64 MERFISH slices. The score>0 (bars in red) and score<0 (bars in cyan) represent enriched and depleted proximity between interacting cell types, respectively. **e** The inference of spatially constrained communication network for GABAergic neurons. Left panel: the cell proximity network. Links in red or gray represent enriched or depleted proximity between interacting cell types, respectively; the width of a link indicates the strength of enrichment or depletion. Middle panel: cell-cell communication network without spatial constraint, calculated based on scRNA-seq data[26] for the same brain region and same cell types (2,044 single cells in total). The width of a link indicates the sum of communication strengths over all significant ligand-target pairs. Right panel: the spatially constrained cell–cell communication network, obtained by removing links with depleted proximity from the original cell-cell communication network. See also Supplementary Fig. 13.

communication networks inferred by NeuronChat, ASD shows not only more total links than control (Fig. 6a, upper panel) but also increased link strength (Fig. 6a, lower panel), indicating an overall enhancement of communications among cell types in ASD. Consistent with this, by contrasting the aggregated intercellular communication networks of ASD and controls, we find that there are more intercellular communications upregulated (Fig. 6b, upper panel) than downregulated (Fig. 6b, lower panel) in ASD patients compared to controls. This may support the hypothesis of local overconnectivity in autism[37–39]. Furthermore, for each cell type and each interaction pair, we calculate the differential outgoing and incoming communication strength in ASD compared to control (Fig. 6c). From the differential outgoing pattern (Fig. 6c, upper panel), we observe most of the glutamate signals are enhanced for most excitatory neuron types; outgoing signals mediated by neuroligin 1 are enhanced for most cell types, but impaired in some cell types such as PV interneuron, oligodendrocyte or OPC (oligodendrocyte precursor cell). From the incoming differential pattern (Fig. 6c, lower panel), we observe that while glutamate signals to most cell types are enhanced, the neuroligin 1 signaling to oligodendrocyte (NRXN1-NLGN1 and NRXN3-NLGN1) and the neuroligin 3 signaling to OPC (NRXN1-NLGN3, NRXN2-NLGN3, and NRXN3-NLGN3) is largely reduced. Interestingly, more than 10 missense mutations in *NLGN3* gene locus have been identified to be associated with ASD, and ASD-associated behavioral phenotypes (such as abnormal social interaction, stereotyped behavior, and enhanced spatial learning) arise in animal models with mutations in *Nlgn3*[40]. Our results suggest that downregulation of *NLGN3* signaling may also underlie dysfunction in ASD. In fact, ASD animal models show reductions in oligodendrocyte numbers and myelination[41], while the

differentiation of OPC to oligodendrocyte is affected by *NLGN3*[42]. Consistent with this evidence, the specific downregulated *NLGN3* signaling to OPC in ASD, discovered in our analysis, suggests that the defective *NLGN3* signaling may cause ASD in a mechanism via the dysfunction of OPC.

We next compare the number of communication links and the information flow for each interaction pair between ASD and controls (Fig. 6d). ASD and controls share similar numbers of links and similar information flow for interaction pairs such as GABA-GABRG3 and NRXN1-NLGN4Y, which may be not specific to ASD. Some interaction pairs such as NRXN2-NLGN2, Glu-GRM3, and Glu-GRIN1, are upregulated in ASD in both number of links and information flow. Many interaction pairs show little difference in the number of links, but with increased information flow, such as NRXN2-NLGN1, Glu-GRIA2 and Glu-GRIA4. Other interactions such as CCK-CCKBR, CRH-CRHR1, TAC1-TACR1 and GABA-GABRA6 show decrease in ASD, in both number of links and information flow.

We further identify the interaction pairs that are conserved or context-specific, by projecting the interaction pairs of ASD and control onto the same two-dimensional manifold and clustering according to their functional similarity (Fig. 6e). All of the ligand-target interaction pairs are categorized into four clusters: cluster 3 and cluster 4 are dominated by GABA signaling and neurexin-neuroligin signaling, respectively; both cluster 1 and cluster 2 are dominated by glutamate signaling, while cluster 1 also includes neuropeptide signaling and gap junction interaction as well as neuroligin 3 signaling. Surprisingly, for most of the interactions, the communication networks for ASD and control are grouped into the same clusters, indicating that most communication patterns are conserved between ASD and control in

terms of functional similarity. For interaction pairs Glu-GRIK1, Glu-GRIK3, Glu-GRIK4, Glu-GRIK5, and Glu-GRIN3A, the communication networks for ASD and control are grouped into cluster 1 and cluster 2, respectively. This means that the senders or receivers for these interaction pairs are different between ASD and control.

## NeuronChat utilizes spatial transcriptomics to infer and visualize neural-specific communication networks

NeuronChat can also be used for the inference and visualization of neural-specific communication networks from spatial transcriptomics (ST) data which measure gene expression in neural cells together with their spatial locations. We illustrate such functionality of NeuronChat by applying it to three mouse brain ST datasets based on three different sequencing techniques including seqFISH+[5], MERFISH[4], and Visium[43]. The seqFISH+ dataset[5] includes mRNA expressions of 10,000 genes in 913 cells in the mouse somatosensory cortex and subventricular zone, where there are 358 excitatory neurons of four types. The MERFISH dataset[4] includes mRNA expressions of 258 genes in approximately 300,000 cells (including nine glutamatergic subclasses and five GABAergic subclasses as well as non-neuronal subclasses) in the mouse primary motor cortex and adjacent areas. The Visium dataset[43] includes mRNA expression profiles in 2,702 spots of a coronal slice of the mouse brain, and these spots are classified into seven clusters.

For all three ST datasets, we compute the communication networks among cell types (or spot clusters) without imposing spatial constraints. For seqFISH+ and Visium datasets, the communication networks are directly calculated from the spatial transcriptomics; for the MERFISH dataset, because the number of genes included in the MERFISH dataset is too small to cover most of the ligand-target pairs, we use the scRNA-seq data of mouse primary motor cortex[26] to infer the communication network among seven excitatory cell types that are shared by the MERFISH dataset and the scRNA-seq dataset (4,461 cells of seven glutamatergic subclasses). To visualize communications networks in space, we develop a multi-layered visualization tool to illustrate together the spatial communications network, cell type/spot cluster annotation, and tissue image/ anatomic reference (Fig. 7a–c; Supplementary Fig. 12).

NeuronChat also provides an option for adding the spatial constraint to the communication network. We test this functionality by applying it to communications among five GABAergic subclasses (i.e., *Lamp5, Sncg, Vip, Sst,* and *Pvalb*) of the MOp cortex, based on the MERFISH[4] (Supplementary Fig. 13) and scRNA-seq data[26]. While long-range spatial communication occurs regularly for neural-specific signals, GABAergic neurons generally have localized axonal arbors and the connection probability among them decreases with interneuronal distance[44]. To study the potential spatial effect on communication networks for GABAergic neurons, we characterize the spatial proximity among GABAergic subclasses by calculating the spatial proximity enrichment score similar to a previous study[45] (Fig. 7d; see Methods). We can then remove communication links with their spatial proximity scores lower than a given threshold, leading to a spatially constrained communication network (Fig. 7e). For example, while *Pvalb* subclass has connections with each of the other four GABAergic subclasses in the communication network without spatial constraint, this subclass only shows enriched cell proximity with itself or *Sst* subclass. This observation is consistent with the evidence that *Pvalb* cells preferably connect to other *Pvalb* cells[46].

## Discussion

NeuronChat is designed specifically for inferring neural-specific intercellular communications from single-cell expression data and spatially resolved transcriptomics. We have constructed a ligand-target interaction database of neural signaling, and presented a computational model that incorporates the process of neural signal

transmission to infer intercellular communications, making NeuronChat different from the existing methods that have been developed for inferring communications among cells for non-neuronal activities. The benchmark and applications of NeuronChat to multiple datasets has shown its ability to predict neural connectivity. Considering the neuron heterogeneity identified from a growing number of single-cell transcriptomic datasets and spatial transcriptomic datasets, novel neural connections among diverse transcriptomic states may be predicted by NeuronChat. By contrasting available neural connectivity data, such as retrograde labeling data and electrophysiological data, with our predicted communication networks for particular ligand-target interaction pairs (e.g., Fig. 2e, j), one can further identify the signaling pathways and key genes that may provide insights into uncharacterized mechanisms underlying neural connectivity.

The NeuronChat R package provides versatile and easy-to-use visualization tools and network analysis approaches, to allow convenient exploration of neural-specific intercellular communication patterns. Using such analysis tool in this study, NeuronChat was shown to classify glutamatergic neuron subtypes and GABAergic neuron subtypes into separate sender groups based on their outgoing signaling patterns; the interaction pairs mediated by glycine are categorized into the interaction group that contains GABA signals from GABAergic neurons; and the specific downregulation in *NLGN3* signaling to OPC has been identified in ASD patients. Collectively, NeuronChat is able to decipher convoluted interneuronal communications with biologically meaningful discoveries from scRNA-seq data. To explore the signaling pathways and gene regulatory networks downstream from the predicted interneuronal communications, one may use the existing database, such as OmniPath[47], STRING[48] and NicheNet[11], to construct integrated cell-cell communication networks through connecting signaling as well as transcriptional regulation.

There is a trade-off between accuracy and computation speed for permutation-based $p$-value calculations, which are needed for all new applications of NeuronChat. A high number of permutations can produce a more accurate empirical $p$-value, but it suffers from long computation time (the time complexity for the permutation-based $p$-value calculations is $O(n)$ where $n$ is the number of permutations). Nevertheless, we find a good consistency between p-values calculated by 100 and 1000 permutations, and the links detected by 1,000 permutations largely overlap with those detected by 100 permutations with no more than 1.5% of links missed (Supplementary Fig. 3). These results suggest that one may reduce the number of permutations (e.g., default number 100 in NeuronChat) to save computation time while maintaining the accuracy of $p$-value calculations.

While NeuronChat's computational workflow has been optimized to predict neuronal connectivity, the settings can be expanded to incorporate more refined models, for example, for estimating the abundance of small molecular neurotransmitters. For such cases, the stoichiometric effects of metabolism and pathways dependency may be included in addition to using expressions of only synthetic enzymes and vesicular transporters. By comparing nine glutamate abundance surrogates, we find that some of the scFEA-derived surrogates show higher AUROC (or AUPRC) values than NeuronChat's ligand abundance (Supplementary Fig. 7). While the difference between NeuronChat's ligand abundance and the best scFEA-derived surrogate is small, it suggests ways in improving the prediction accuracy of neuronal connectivity.

Like other existing methods for inferring cell-cell communications, NeuonChat estimates the abundance of ligands and target proteins from transcriptomics that could be inconsistent with protein or metabolite levels. In principle, NeuronChat can be applied to proteomics and metabolomics data to infer ligand-target interactions if the data becomes available. With the single-cell proteomics and metabolomics techniques lagging behind transcriptomics in coverage

of molecules or throughput[49,50], for now the transcriptomics data remain as a main data source for cell-cell communication inference.

While NeuronChatDB includes major small-molecular neurotransmitters, most of the neuropeptides, some gasotransmitters, gap junction proteins as well as synaptic adhesion molecules, there may be missing information in the curated interaction entries, leading to bias in the inference. Nevertheless, NeuronChat allows easy updating of the database with user-defined interactions that are not included in the current version, to expand its applicability for more interactions.

In the current study, NeuronChat splits the multiple different subunits of the heteromeric receptor into separate entries when evaluating their abundance. In principle, the subunit stoichiometries need to be taken into account to more accurately represent the heteromeric receptors. The heteromeric receptor can be assembled from various combinations of subunits, with great diversity in subunit compositions[51], dramatically affecting its functional properties. For example, the existence of the GluA2 subunit in AMPA receptors determines the permeability to calcium ions[52]. While NeuronChatDB does not contain the information of subunit stoichiometries that are largely unknown for most heteromeric receptors, NeuronChat includes the option to model the abundance of particular heteromeric receptors with customer-provided subunit stoichiometries. We expect that incorporating knowledge of auxiliary proteins and downstream genes will improve the accuracy of the communication prediction.

## Methods

### Database construction for ligand-target interactions

NeuronChatDB is curated from existing databases (including KEGG[53] and IUPHAR/BPS Guide to PHARMACOLOGY[54]) and literature (e.g., neuropeptide interactions are from the reference[16]), and contains neural-specific intercellular molecular interactions for both mouse and human. There are 373 entries in total. Each entry of NeuronChatDB represents an interaction pair, including one ligand and a cognate target as well as genes related to them. The ligands include small-molecule neurotransmitters, neuropeptides, gap junction proteins, gasotransmitters and synaptic adhesion molecules: small-molecule neurotransmitters include glutamate (Glu), GABA, glycine (Gly), acetylcholine (ACh), serotonin (5-HT), dopamine (DA), epinephrine (Epi) and norepinephrine (NE); gasotransmitters include carbon monoxide (CO) and Nitric oxide (NO); synaptic adhesion molecules refer to neurexins (regarded as the ligand) and neuroligins (regarded as the target). The targets are typically but not limited to receptors. For example, the target proteins for neurotransmitters can also be uptake transporters or deactivating enzymes; the target proteins for gap junction proteins are other compatible gap junction proteins. For non-peptide neurotransmitters, corresponding synthesizing enzymes and/or vesicular transporters are included in the entry; for heteromeric receptors that contain multiple different subunits, corresponding subunits are split into different entries with the same ligand. To be compatible with the inference model of NeuronChat, for the non-peptide neurotransmitters, related genes including vesicular transporters and synthesizing enzymes responsible for different catalyzing steps are annotated into separate groups.

### Inference of neural-specific cell-cell communications

For each ligand-target interaction pair, NeuronChat infers intercellular communication in three steps as follows:

1. Calculation of ensemble average expression. For each gene involved in the ligand-target interaction pair, the ensemble average expression in a given cell group is calculated using Tukey's trimean:

$$TM = \frac{1}{2}Q_2 + \frac{1}{4}(Q_1 + Q_3) \qquad (1)$$

where $Q_1$, $Q_2$, and $Q_3$ are the first, second and third quartile of the expression levels of the gene in the given cell group. Because the genes expressed in less than 25% of cells are filtered out, Tukey's trimean benefits to identify the cell-type enriched ligand-target pairs (see also Supplementary Fig. 8).

2. Calculation of cell-cell communication strength. NeuronChat estimates the abundance of the ligand and the target for each cell group, and computes cell-cell communication strength. When the ligand is a peptide or protein that corresponds to a single gene, the abundance of the ligand for a given cell group is set as the ensemble average expression defined in step 1. For non-peptide neurotransmitters, the abundance of the ligand depends on expression levels of corresponding synthesizing enzymes and vesicular transporters. Assume that the synthesis of the ligand requires $m_1$ catalyzing steps; for the $s$-th catalyzing step ($s = 1, 2, \ldots, m_1$), let $p_s$ denote the number of isoenzymes that catalyze the same chemical reaction (e.g., glutamate decarboxylase 1 and 2 for the synthesis of GABA), and $E_{i,s,l}$ ($l = 1, 2, \ldots, p_s$) denote the ensemble average expression of the $l$-th isoenzyme for step $s$ in cell group $i$. Likewise, let $q$ denote the number of vesicular transporters for the storage of the same ligand (e.g., vesicular glutamate transporter 1, 2, and 3 for the glutamate), and let $V_{i,l}$ ($l = 1, 2, \ldots, q$) denote the ensemble average expression of the $l$-th vesicular transporter. Then, the abundance of ligand is modeled by the $1 + m_1$ functional groups of genes including one group for vesicular transporters and $m_1$ groups for the $m_1$ steps of synthesis. Because a high abundance of ligand requires high expressions of all the $1 + m_1$ groups of genes, so the AND logic (i.e., geometric mean) is applied among different groups of genes; since the genes within the same group are redundant for the same function, the OR logic (i.e., arithmetic mean) is applied. Thus, the abundance of ligand for the $i$-th cell group is modeled as

$$L_i = \sqrt[1+m_1]{\frac{\sum_{l=1}^{q} V_{i,l}}{q} \cdot \frac{\sum_{l=1}^{p_1} E_{i,1,l}}{p_1} \cdots \frac{\sum_{l=1}^{p_{m_1}} E_{i,m_1,l}}{p_{m_1}}}. \qquad (2)$$

When some of the redundant genes for a function group are missing from the input data due to low gene coverage, NeuronChat will only use the remaining genes. If the entire group of genes are missing, the ligand abundance is set to be zero by default; in such case, NeuronChat also provides a less-strict mode to allow the calculation by setting the ensemble average expressions of these genes as ones, which will be useful for the dataset with low gene coverage.

In the current study, all targets correspond to a single gene (for heteromeric receptors that contain multiple different subunits, we just split these subunits into different entries but with the same ligand so that the target for each entry is represented by a single subunit), so the target abundance $T_j$ for the cell group $j$ is set as the ensemble average expression of the corresponding gene defined in step 1). Nevertheless, NeuronChat is also compatible with heteromeric receptors given customer-provided subunit stoichiometries, and the target abundance in cell group $j$ is defined as the weighted geometric mean of the ensemble average expressions of subunit genes:

$$T_j = \sqrt[\sum_{l=1}^{m_2} c_l]{T_{j,1}^{c_1} \cdots T_{j,m_2}^{c_{m_2}}} \qquad (3)$$

where $c_l's$ ($l = 1, 2, \ldots, m_2$) are the subunit stoichiometries and $m_2$ is the number of different subunits. Then the communication strength from cell group $i$ to cell group $j$ is defined as

$$LT_{i,j} = L_i \cdot T_j \qquad (4)$$

3. Determination of the significance of communication links. The statistical significance of each communication is calculated through the permutation test by randomly permuting group labels of cells and then recalculating the communication strength for each permutation. Then the *p*-value is calculated as

$$p_{i,j} = \frac{1}{M} \sum_{m=1}^{M} I_{\left\{ LT_{i,j}^{(m)} > LT_{i,j} \right\}} \tag{5}$$

Where $LT_{i,j}^{(m)}$ is the recalculated ligand-receptor interaction score for the *m*-th permutation and *M* is the total number of permutations ($M = 100$ by default); $I_{\{LT_{i,j}^{(m)} > LT_{i,j}\}}$ is an indicator function of *m*, and equals 1 if $LT_{i,j}^{(m)} > LT_{i,j}$ and 0 otherwise. The *p*-value is corrected for multiple tests by using Benjamini–Hochberg procedure and communications with *p*-value < 0.05 are considered significant; for non-significant communication links, the communication strength values are set to be zeros.

By performing the steps 1–3 for each ligand-target interaction pair, we obtain the communication strength for any interaction pair *k* from any cell group *i* to cell group *j*, $P_{i,j}^k$ ($i = 1, \ldots, G, j = 1, \ldots, G, k = 1, \ldots, K$), which can be written into a three-dimensional array $\mathbf{P}$ ($G \times G \times K$), where *G* is the number of cell groups and *K* is the number of ligand-target interaction pairs.

### Implementation of CellChat and CellPhoneDB
We use the same computational workflow of NeuronChat for the implementation of CellChat and CellPhoneDB, except for the calculation of ligand abundance and the formula for the communication strength: CellChat computes the ligand abundance as the geometric mean of the average expressions of genes contributing to ligand emission, and adopts a Hill function to transform the product of ligand and target abundance to get the communication strength; CellPhoneDB computes the ligand abundance as the minimum of the average expressions of contributing genes, and takes the mean of the ligand and target abundance as the communication strength. It should be noted that, for small-molecule neurotransmitters, both of CellChat and CellPhoneDB are implemented without categorizing the contributing genes into different functional groups and thus use AND logic (geometric mean or minimum) for all contributing genes, without applying OR logic for redundant genes for the same function; hence, the ligand abundance can be dramatically underestimated if some of the redundant genes are expressed at extremely low levels (e.g., zeros).

### Inference of neural-specific cell-cell communications from spatial transcriptomics data
For spatial transcriptomics data, NeuronChat can calculate the communication networks without imposing spatial constraints, by using the cell-by-gene (or spot-by-gene) count matrix. To construct the spatially constrained communication networks, we first use the spatial locations of cells/spots to characterize the proximity among cell groups/spot clusters by calculating spatial proximity enrichment score similar to Giotto[45], and then remove communication links (from the communication network without spatial constraint) with their spatial proximity scores lower than a given threshold. Specifically, for the calculation of spatial proximity enrichment score, we first find all cell/spot pairs within a given distant threshold (400 microns used for the MERFISH dataset), and calculate the observed frequencies for all combinations of cell group/spot cluster pairs. Then we randomly permutate cell/spot labels to recalculate the frequencies for cell group/spot cluster pairs, and the expected frequencies are obtained by averaging the figures over 1000 permutations by default. The spatial

proximity enrichment score for two cell groups/spot clusters is calculated as the log2-transformed ratio of the observed frequency over the expected frequency; a high spatial proximity enrichment score means the two cell groups/spot clusters are preferentially located close to each other in space. The associated *p*-value is calculated as the percentage of permutations that yield frequency values higher than the observed one.

### Methods used for aggregating the intercellular communication networks over all interaction pairs
For each interaction pair *k*, denote $\mathbf{P}^{(k)} = (P_{i,j}^k)$ the communication strength matrix for the significant links among all cell groups (the values for non-significant communication links are set to be zeros). The aggregated cell-cell communication network can then be obtained by summarizing all $\mathbf{P}^{(k)}$ over all interaction pairs, using one of the four aggregation methods:

1. $\mathbf{C}_1 = \sum_k \mathbf{P}^{(k)}$. Given the sending and receiving cell group, this aggregation method sums communication strength values over all interaction pairs, denoted as "weight".
2. $\mathbf{C}_2 = \sum_k \mathbf{I}^{(k)}(a_k)$ where $\mathbf{I}^{(k)}(a_k)$ is a matrix with the same dimensions as $\mathbf{P}^{(k)}$ and its element $I_{i,j}^k(a_k) = 1$ if $P_{i,j}^k > a_k$ and $I_{i,j}^k(a_k) = 0$ otherwise. If $a_k = 0$ for any *k*, then the element in $\mathbf{C}_2$ counts the number of links with non-zero communication strength over all interaction pairs, from one cell group to another; in such case, the aggregation method is denoted as "count".
3. $\mathbf{C}_3 = \sum_k w_k \mathbf{I}^{(k)}(a_k)$, where $w_k$ is the sum of elements in $\mathbf{P}^{(k)}$ and also known as the information flow for the interaction pair *k*. If $a_k = 0$ for any *k*, this aggregation method counts the number of links with non-zero communication strength while assigning the weight of the interaction pair as the information flow, denoted as "weighted count".
4. $\mathbf{C}_4 = \sum_k \mathbf{P}^{(k)} \odot \mathbf{I}^{(k)}(a_k)$, where $\odot$ means Hadamard product, i.e., $(\mathbf{A} \odot \mathbf{B})_{i,j} = A_{i,j} B_{i,j}$. The aggregation method, denoted as "thresholded weight", sums the communication strength values over all interaction pairs with the communication strength values filtered by a threshold for each interaction pair. This method is the default setting for benchmarking NeuronChat, with threshold $a_k$ defined as the 80% quantile of all the elements of $\mathbf{P}^{(k)}$.

### Network analysis approaches
Network analysis approaches include pattern recognition[32,33] and manifold and classification learning[35], the implementation of which is based on CellChat functions[9]. For pattern recognition, the latent patterns for the outgoing signaling of sending cells (or incoming signaling of receiving cells) are calculated through the non-negative matrix factorization of a two-dimensional matrix obtained by summing the three-dimensional array $\mathbf{P}$ along the second (or first) dimension. The two output matrices, cell loading matrix and signaling loading matrix, which represent the correspondence between cell groups and latent patterns and the correspondence latent patterns between individual interaction pairs, respectively, can be visualized by the alluvial plots. The manifold and classification learning projects the communication networks for individual interaction pairs into a low-dimensional manifold and classifies them into groups. The first step is the calculation of the functional similarity matrix, of which each element is defined as the ratio of the number of overlapped communication links to that of non-overlapped links for two communication networks. Then *k*-nearest neighbors are found for each interaction pair based on the functional similarity matrix and used to smooth the functional similarity matrix. Finally, the smoothed similarity matrix is used to perform uniform manifold approximation and projection (UMAP) and the interaction pairs are grouped based on the *k*-means clustering of the first two components of the learned manifold.

## Data preprocessing

The single-cell RNA count matrices were processed as follows, prior to analysis with NeuronChat. The RNA counts for each cell were divided by the total counts in the cell and multiplied by a scale factor (10,000 by default), and these values are added with a pseudocount of 1 and then natural-log transformed. For the genes related to interacting molecules in NeuronChatDB and used to calculate the communication strength, the expression values are further normalized by the maximum, to guarantee the communication strength has the range from 0 to 1.

## Reporting summary

Further information on research design is available in the Nature Portfolio Reporting Summary linked to this article.

## Data availability

NeuronChatDB is included in the NeuronChat repository (https://github.com/Wei-BioMath/NeuronChat), and can be also accessed in table formats (https://github.com/Wei-BioMath/NeuronChatAnalysis2022/tree/main/NeuronChatDB_table). KEGG pathway database is available at https://www.genome.jp/kegg/pathway.html. IUPHAR/BPS Guide to PHARMACOLOGY database is available at https://www.guidetopharmacology.org. The mouse scRNA-seq datasets analyzed in this study are available from the Gene Expression Omnibus (GEO) repository under the following accession numbers: GSE115746 and GSE185862. The human autism datasets analyzed in this study are available at https://autism.cells.ucsc.edu. The processed seqFISH+ data are available at https://rubd.github.io/Giotto_site/articles/mouse_seqFISH_cortex_200914.html. The MERFISH dataset is available at https://doi.org/10.35077/g.21. The Visium dataset is available at https://support.10xgenomics.com/spatial-gene-expression/datasets/1.1.0/V1_Adult_Mouse_Brain. The code used to reproduce the figures and results in this study are available at https://github.com/Wei-BioMath/NeuronChatAnalysis2022.

## Code availability

NeuronChat is an R package available at GitHub (https://github.com/Wei-BioMath/NeuronChat) and Zenodo (https://doi.org/10.5281/zenodo.7600421)[55]. The package dependencies include data.table v1.14.2, dplyr v1.0.9, CellChat v1.1.3, NMF v0.23.0, Seurat v4.1.0, SeuratObject v4.1.0, circlize v0.4.14, ComplexHeatmap v2.8.0, igraph v1.3.4, ggalluvial v0.12.3 and ggplot2 v3.3.6. The code used to reproduce the analysis in this study is available at https://github.com/Wei-BioMath/NeuronChatAnalysis2022.

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

## Acknowledgements

This work is supported in part by a BRAIN Initiative Grant (NS104897) from the US National Institutes of Health, a grant from National Institute on Aging (U01AG076791), an NSF grant DMS1763272 (Q.N.) and a Simons Foundation grant (594598, Q.N.).

## Author contributions

Q.N. and W.Z. conceived this work; W.Z. implemented the method; W.Z. and H.R. performed the simulations, and prepared figures; W.Z., K.G.J., X.X., and Q.N. interpreted the results; W.Z. and Q.N. wrote the paper; all authors edited and approved the final manuscript.

## Competing interests

The authors declare no competing interests.
