## [Peer Review File · Nature Communications]

Inferring neuron-neuron communications from single-cell transcriptomics through NeuronChatREVIEWER COMMENTS

Reviewer #1 (Remarks to the Author):

The communications between neurons play critical roles in our central nervous system. Therefore, understanding the interactions between neuron cells is essential for understanding how our brain functions. Single-cell genomics provides unrivaled opportunities to study the interactions between different cells. In the past years, numerous computational methods, such as cellphonedb and cellchat, have been developed to infer the communications between cells. Nevertheless, none of these methods, at least to my knowledge, considers the uniqueness of neural communication. The authors filled this gap by providing one of the very first tools to infer communications between neuron cells.

The strength of the study:

- 1) The authors built/curated one of the first neural-specific database of intercellular interactions for both humans and mice, which was named NeuronChatDB. This dataset can surely facilitate other works on studying the interactions between neurons by providing the "ligand-target dictionary" required to infer neural interactions, and thus I believe that this newly developed tool can substantially benefit the neurobiology field.
- 2) Because the interactions between neuron cells are distinct from the interaction between other cells, the authors developed a designated method to infer the interactions between neurons. For the non-peptide neurotransmitters, the ligand abundance calculation takes the expressions of corresponding synthesizing enzymes and vesicular transporters into consideration and thus is more complicated than the simple Ligand-Receptor based "cross-talking" scoring scheme employed by cellphonedb, cellchat, and many other methods.
- 3) Similar to the previous work from the authors (cellchat), the authors also provide very comprehensive visualizations of the results from the model, which will undoubtedly facilitate the adaptation of the methods by the community (similar as cellchat) and benefit various neurobiology studies.

Although I am convinced about the potential impact and popularity of the method, I have a few comments that should be addressed.

Major comments:

1)As single-cell spatial data is becoming more and more abundant, the author should consider extending their method to incorporate spatial information for the inference of neural cell interactions. Undoubtedly, the spatial information of all the neuron cells could help better infer the interactions. For example, closer cells tend to be more interactive compared to distant cells (given the expression of the enzymes and vesicular transporters are similar or identical).

2)In this manuscript, the author did not benchmark their method (neuronchat) with other methods. I do understand that there might not exist any other methods that were designed specifically to infer the interactions between neurons. The authors also compared the prediction of their method to the ground truth obtained from retrograde labeling. However, I am wondering whether the authors could show whether a simple extension of the existing methods (say, the cellchat developed by the authors a few years ago) could also provide a good cell-cell interaction inference. Ofcourse, the "Ligand-target" database (neuronchatDB) is still required as the L-R reference for those existing methods. This comparison (particularly if a simple extension fails to achieve a good performance) can further demonstrate the need for a new method (i.e., Neuronchat). The author did explain the limitation of existing methods in the introduction "all existing methods are based on short-range autocrine/paracrine signaling which only acts through ligand diffusion and physical contact of cells ."

While I agree with the authors that the neural interactions could be more distant, existing methods might still work well because of their simplicity. For example, cellphonedb actually did not take the "range" into the calculation of the interaction score. For a given Ligand-Receptor pair L-R, the method calculates the interaction score by taking the minimum value of the average ligand (of the sender cluster) and receptor (of the receiver cluster). The distances between the cells are not considered in the model, such a simplified calculation may also be able to circumvent the limitations and capture the interactions between distant neuron cells. Therefore, a direct comparison will provide a much more vigorous justification. T

Minor comments:

1) "retrograde labeling", can the authors provide one or two sentences to describe the retrograde labeling and how it profiles the real cell-cell interactions?

2)Can authors provide a justification of using Tukey's trimean, why not just the simple mean of the gene expression in all cells of a group? What is the disadvantage of doing this? Why could the trimean approach resolve the issue?

3)Please rephrase "Assume that the synthesis of the ligand requires m_1 stepsThus, the abundance of ligand for the i -th cell group is modeled as". It's not easy to read and understand.

4) One potential limitation of those permutation-based p-value calculations is that it is very time-consuming. Are the random permutations needed for all new applications (data)? If so, what is the time complexity? The empirical p-value accuracy depends on the # of permutations. A high # of permutations can present a more accurate empirical p-value, but it will suffer from long computation time. On the other hand, if we reduce # of permutations, the p-value calculation loses its accuracy.

Reviewer #2 (Remarks to the Author):

(A MORE READABLE PDF VERSION IS ATTACHED BELOW!)

This manuscript presents NeuronChat, a novel package for predicting and visualizing neuron-type-specific communication networks from neurotaxonomically classified single-cell RNA-seq expression data. The concept here, and as pioneered by previous studies (e.g., cited Ref. #36), is to use products of mRNA abundance metrics from ligand-receptor pairs as predictions cell-cell signaling between pairs of transcriptomically defined cell types. The method is introduced in the present work making use of a mouse cortex neurotaxonomy and genome-wide scRNA-seq datasets from cited Ref. #3 and then demonstrated by applications to human control/autism neurotaxonomies from scRNA-seq datasets from Ref. #22. Given the very large numbers of cell types and ligand-receptor pairs recognized today in virtually all metazoan brains, improved abilities to develop specific and testable hypotheses regarding communications between specific cell types seem certain to advance neuroscience.

A major strength of the present work is thus development of a computational package for visualizing and exploring such predictions. Unfortunately, the presentation as it stands lacks clarity at some critical junctures and leaves room for doubt as to the strength and accuracy of the "inferences" claimed as results from NeuronChat. The lack of presentation clarity is most severe in the critical section of Results entitled "Benchmarking of NeuronChat". Noting the many "false positives" evident in Figs. 2b and 2g, juxtaposed

with the high incidence of "true positives" in these graphs, it is hard to see support for the statement that

"NeuronChat is able to predict neural connectivity accurately and robustly". (...particularly the part about "accurately"...). This is particularly so given that products of numerous glutamate receptor genes are expressed in virtually all CNS neurons and that Cck and Cckbr transcripts are abundant in almost all cortical

glutamatergic neurons. We may be missing something here, but we fear that many other readers may have a problem with this critical passage. This section could use a very substantial re-write aimed either at more clearly leading the reader to the stated conclusion and/or softening the conclusion of "accuracy".

Another major issue is that the precise workflow is not clearly illustrated, and Figure 1 presents this workflow only in very general terms. While Figure 1 is attractive, a more detailed illustration of how interactions are represented, how gene expression is used to infer connection strength, and at least an indication of the computational model approach would help. It is difficult to understand this workflow, except at a very conceptual level, through initial text in the manuscript. For example, in the annotation of these interactions, is this a strictly binary interaction or does the data base indicate interaction strength

in any way? This additional clarification to the manuscript is important as that while considerable methods are developed for the statistical methodology, the overall workflow must be first rigorously presented.

The quantification model of the gene expression interactions is novel in many ways and a strength of the paper, and the permutation testing framework is an important addition to this approach. The analysis and comparison of the VISp and ALM datasets from the Allen Institute is a strength of the manuscript and

offers some interesting communication links for follow up. The analysis in this part of the manuscript is generally rigorously accomplished, however, methods used for aggregating intercellular communication should be prioritized as to which are the best approaches the methods do not make this clear. It also seems likely that caveats about potential disconnects between transcript metrics and abundance of proteins or their enzymatic products. Additionally, the molecular interactions database is quite small at 373 entries and the results of the present work may be biased by this size restriction.

Though the NeuronChat software package potentially offers considerable value, the present documentation falls short of enabling its effective use. The software package itself does not completely meet the standards proposed in the manuscript. There are a large number of installation dependencies in the code and these reviewers were unable to fully resolve them with the present R version. It would be most helpful to have a short example tutorial in illustrating one or more of the results of the manuscript, beyond the code the figures. If the clarity of presentation and software documentation can be improved as we suggest below and the claims of accuracy can be more clearly supported or else attenuated appropriately, NeuronChat has the potential to evolve into a valuable increment in the cellular neuroscientist's toolkit.

Finally, while we are enthusiastic about the package and approach the value of NeuronChat might be further enhanced and clarified by citing some precedents within its neuroscientific scope. One of these is already cited as Ref. #36, but the current citation only references one table that pairs neuropeptide precursors and their receptors. The authors should note that this publication is specifically about network predictions from scRNA-seq data made according to the same general logic that is at the heart of NeuronChat and directly based on the same Allen Institute neurotaxonomy and dataset as much of work presented in the current manuscript. Citation of two additional previous publications would be appropriate

1. Smith SJ, Hawrylycz M, Rossier J, Sumbul U: New light on cortical neuropeptides and synaptic network plasticity. *Curr Opin Neurobiol* 2020, 63:176-188. PMID: 32679509
2. Smith SJ: Transcriptomic evidence for dense peptidergic networks within forebrains of four widely divergent tetrapods. *Curr Opin Neurobiol* 2021, 71:100-109. PMID: 34775262

These two publications both make neurotaxonomic/transcriptomic network predictions based on the same ligand-receptor products as the present manuscript, but also point toward the necessity of prediction testing and relevant experimental and phylogenomic tests. In summary, this work represents an advance in a comparatively new area of quantifying interaction networks from single cell expression data. While we remain enthusiastic, the manuscript should be improved as indicated to make NeuronChat

more fully useable and indicate its contribution to an exciting research approach to cell network

communication.

Reviewer #3 (Remarks to the Author):

In the manuscript “Inferring neuron-neuron communications from single-cell transcriptomics through NeuronChat”, Zhao et al. adopted a conventional cell-cell communication framework to make a specialized methodology for inferring neuron-neuron communication based on scRNA-seq data. More precisely, the method is built upon the tool CellChat which was developed and published by the same lab a while back. The authors named their new tool NeuronChat. From an application standpoint, like CellChat, which is one of the most popular tools for cell-cell communication analysis, NeuronChat is a well-documented R package, with clear tutorials accompanied. Both CellChat and NeuronChat generate very nice visualization. I do think NeuronChat is a great piece of software extension. However, almost all analysis in NeuronChat, such as latent patterns, functional similarity, have already been highlighted in the original CellChat paper. The methodological details explained in the methods section look very similar to CellChat. I feel that from a method development standpoint, with respect to CellChat, the novelty of NeuronChat is rather incremental.

The authors emphasized that neural communication is mediated by neurotransmitters, which are non-peptides and thus excluded from most existing ligand-receptor databases. The authors therefore curated a list of neuron-specific ligand-receptors, which are not included in CellChat. Their efforts are appreciated. Nevertheless, the abundance of small molecules cannot be directly measured in scRNA-seq data, authors therefore used the expression of the related enzymes as a proxy. While the proxy makes sense, they are also quite the obvious choice. Stoichiometric effects of metabolites and pathways dependency are not considered. There are a few recently published methods that leverage single-cell RNA-seq data to estimate metabolic flux, for instance, the tool scFEA (<https://genome.cshlp.org/content/31/10/1867>), and therefore go beyond the simple proxy used by NeuronChat.

As the only quantitative justification of NeuronChat, the authors used two projective networks identified using retrograde labeling as the gold standard. Even though a reference is provided, it seems that the details of the two networks were not mentioned. I assume retrograde labeling measure connections in a single-cell level, but NeuronChat predicts connections in cell-type level. It is not clear how sensitivity and specificity are defined. As the topology of the gold standard networks might affect the results (for instance, a cell-type is connected to many other cell-type, making prediction easier), the current AUC values (0.83, 0.76 etc.) will make more sense if the authors could repeat the analysis but shuffling the edges in the gold standard networks. I wonder if the resultant AUCs would reduce to 0.5. If not, I am not sure how to interpret the current AUC values, which are already not too impressive.

Response to Reviewer #1

The communications between neurons play critical roles in our central nervous system. Therefore, understanding the interactions between neuron cells is essential for understanding how our brain functions. Single-cell genomics provides unrivaled opportunities to study the interactions between different cells. In the past years, numerous computational methods, such as cellphonedb and cellchat, have been developed to infer the communications between cells. Nevertheless, none of these methods, at least to my knowledge, considers the uniqueness of neural communication. The authors filled this gap by providing one of the very first tools to infer communications between neuron cells.

The strength of the study:

1) The authors built/curated one of the first neural-specific database of intercellular interactions for both humans and mice, which was named NeuronChatDB. This dataset can surely facilitate other works on studying the interactions between neurons by providing the "ligand-target dictionary" required to infer neural interactions, and thus I believe that this newly developed tool can substantially benefit the neurobiology field.

2) Because the interactions between neuron cells are distinct from the interaction between other cells, the authors developed a designated method to infer the interactions between neurons. For the non-peptide neurotransmitters, the ligand abundance calculation takes the expressions of corresponding synthesizing enzymes and vesicular transporters into consideration and thus is more complicated than the simple Ligand-Receptor based "cross-talking" scoring scheme employed by cellphonedb, cellchat, and many other methods.

3) Similar to the previous work from the authors (cellchat), the authors also provide very comprehensive visualizations of the results from the model, which will undoubtedly facilitate the adaptation of the methods by the community (similar as cellchat) and benefit various neurobiology studies.

Although I am convinced about the potential impact and popularity of the method, I have a few comments that should be addressed.

Response: We are glad the reviewer finds the potential impact of our method. We thank the reviewer for the valuable comments. Below we provide detailed responses to each specific comment.

Major comments:

1. As single-cell spatial data is becoming more and more abundant, the author should consider extending their method to incorporate spatial information for the inference of neural cell interactions. Undoubtedly, the spatial information of all the neuron cells could help better infer the interactions. For example, closer cells tend to be more interactive compared to distant cells (given the expression of the enzymes and vesicular transporters are similar or identical).

Response: Thank you for the nice suggestion. In the revision, we have added new functionality in two ways: using spatial data for inference of cell-cell communication networks and a new multilayered visualization of spatial cell-cell communication (new Figure 7), in an added new section “NeuronChat utilizes spatial transcriptomics to infer and visualize neural-specific communication networks”.

Specifically, we have shown the added functionality using three spatial transcriptomics datasets based on three different sequencing techniques including seqFISH+, MERFISH, and Visium. The seqFISH+ dataset includes mRNA expressions of 10,000 genes in 913 cells in the mouse somatosensory cortex and subventricular zone, where there are 358 excitatory neurons of four types. The MERIFH dataset includes mRNA expressions of 258 genes in approximately 300,000 cells (including nine glutamatergic subclasses and five GABAergic subclasses as well as non-neuronal subclasses) in the mouse primary motor cortex and its adjacent areas. The Visium dataset includes mRNA expression

profiles in 2,702 spots of a coronal slice of the mouse brain, and these spots are classified into seven clusters. For all three ST datasets, we computed the communication networks among cell types (or spot clusters) without imposing spatial constraints. The neural-specific signals can transmit over long spatial distances through various physical connections among neurons that may locate far apart. To better visualize spatial communication, we have developed a new multilayered visualization functionality to illustrate together the spatial communications network, cell type/spot cluster annotation, and tissue image/ anatomic reference (new Figures 7a-7c).

To study the potential spatial effect on communication networks, next we have characterized the spatial proximity among cell types by calculating spatial proximity enrichment score similar to a previous study (Giotto). Using this information, we can remove communication links with their spatial proximity scores lower than a given threshold. Since GABAergic neurons generally have localized axonal arbors and the connection probability among them decreases with inter-neuronal distance, we showcase this functionality by applying it to communications among five GABAergic subclasses (i.e., *Lamp5*, *Sncg*, *Vip*, *Sst*, and *Pvalb*) of the MOp cortex (new Figures 7d and 7e). This spatial constraint functionality has been implemented as a user option in NeuronChat.

These results have been added on Pages 18-20.

2. In this manuscript, the author did not benchmark their method (neuronchat) with other methods. I do understand that there might not exist any other methods that were designed specifically to infer the interactions between neurons. The authors also compared the prediction of their method to the ground truth obtained from retrograde labeling. However, I am wondering whether the authors could show whether a simple extension of the existing methods (say, the cellchat developed by the authors a few years ago) could also provide a good cell-cell interaction inference. Of course, the "Ligand-target" database (neuronchatDB) is still required as the L-R reference for those existing methods. This comparison (particularly if a simple extension fails to achieve a good performance) can further demonstrate the need for a new method (i.e., Neuronchat). The author did explain

the limitation of existing methods in the introduction "all existing methods are based on short-range autocrine/paracrine signaling which only acts through ligand diffusion and physical contact of cells."

While I agree with the authors that the neural interactions could be more distant, existing methods might still work well because of their simplicity. For example, cellphonedb actually did not take the "range" into the calculation of the interaction score. For a given Ligand-Receptor pair L-R, the method calculates the interaction score by taking the minimum value of the average ligand (of the sender cluster) and receptor (of the receiver cluster). The distances between the cells are not considered in the model, such a simplified calculation may also be able to circumvent the limitations and capture the interactions between distant neuron cells. Therefore, a direct comparison will provide a much more vigorous justification.

Response: We thank the reviewer for the good suggestion. To demonstrate NeuronChat's capability in identifying neural-specific communications, in the revision we have added comparisons with CellChat and CellPhoneDB in predicting neuronal connectivity using the same ligand-target database (new Figure 3).

Specifically, we use the same computational workflow of NeuronChat for the implementation of CellChat and CellPhoneDB, except for the calculation of ligand abundance and the formula for communication strength (Please see more details added in Method for comparison (Page 27)). On the inference of neuronal connectivity in both VISp and ALM projection networks, we have shown that NeuronChat outperforms existing cell-cell communication inference methods: 1) the aggregated communication network for NeuronChat has the highest AUROC/AUPRC among the three benchmarking methods (new Figures 3a-3b for VISp and new Figures 3f-3g for ALM); 2) for individual communication networks, NeuronChat not only detects more significant interaction pairs but also yields higher AUROC and AUPRC than the other two methods (new Figures 3c-3e for VISp and new Figures 3h-3j for ALM). These results have been added on Pages 9-10.

The Minor comments:

1. *"retrograde labeling", can the authors provide one or two sentences to describe the retrograde labeling and how it profiles the real cell-cell interactions?*

Response: In the revision, we add a description of retrograde labeling and how it identifies neuronal connectivity on Pages 5-6 as follows: "The connections from excitatory neurons of VISp and ALM to their cortical target regions were identified using monosynaptic retrograde labeling³, where the viral tracers are injected into target regions and move towards the presynaptic neurons via retrograde axonal transport without further spreading to indirectly contacted cells, allowing the identification of direct neural connections²¹⁻²⁵. By grouping retrogradely labeled neurons using their cell type annotations, we obtain the coarse-grained projection networks composed of directed links from excitatory neuron types in VISp and ALM to their cortical target regions (Figure 2a for VISp and Figure 2f for ALM), which are then used for subsequent benchmarking."

2. *Can authors provide a justification of using Tukey's trimean, why not just the simple mean of the gene expression in all cells of a group? What is the disadvantage of doing this? Why could the trimean approach resolve the issue?*

Response: In the revision, we have now made a comparison between Tukey's trimean and arithmetic mean in inferring communications with detailed descriptions on the difference between the two methods (new Supplementary Figure 8).

According to their definitions, for a given gene and a given cell group, the non-zero Tukey's trimean only occurs if the gene is expressed in at least 25% of cells while non-zero arithmetic mean occurs if the gene is expressed in at least one cell. Because the genes only expressed in a small proportion (less than 25%) of cells are filtered out, Tukey's trimean benefits to identify the cell-type enriched ligand-target pairs. As expected, Tukey's trimean leads to fewer detected interaction pairs than arithmetic mean (Supplementary Figures 8a and 8e); however, the interaction pairs produced by Tukey's

trimean show overall higher AUROC and AUPRC than those produced by arithmetic mean, suggesting Tukey's trimean is able to infer more reliable interaction pairs (Supplementary Figures 8b-8d and 8f-8h).

In the revision, we have added this result on Page 11, and detailed descriptions for Tukey's trimean on Pages 24-25.

3. Please rephrase "Assume that the synthesis of the ligand requires m_1 steps Thus, the abundance of ligand for the i -th cell group is modeled as". It's not easy to read and understand.

Response: Sorry for the confusion. In the revision, we have rewritten this paragraph on Page 25 as follows: "Assume that the synthesis of the ligand requires m_1 catalyzing steps; for the s -th catalyzing step ($s = 1, 2, \dots, m_1$), let p_s denote the number of isoenzymes that catalyze the same chemical reaction (e.g., glutamate decarboxylase 1 and 2 for the synthesis of GABA), and $E_{i,s,l}$ ($l = 1, 2, \dots, p_s$) denote the ensemble average expression of the l -th isoenzyme for step s in cell group i . Likewise, let q denote the number of vesicular transporters for the storage of the same ligand (e.g., vesicular glutamate transporter 1, 2, and 3 for the glutamate), and let $V_{i,l}$ ($l = 1, 2, \dots, q$) denote the ensemble average expression of the l -th vesicular transporter. Then, the abundance of ligand is modeled by the $1 + m_1$ functional groups of genes including one group for vesicular transporters and m_1 groups for the m_1 steps of synthesis. Because a high abundance of ligand requires high expressions of all the $1 + m_1$ groups of genes, so the AND logic (i.e., geometric mean) is applied among different groups of genes; since the genes within the same group are redundant for the same function, the OR logic (i.e., arithmetic mean) is applied. Thus, the abundance of ligand for the i -th cell group is modeled as..."

4. One potential limitation of those permutation-based p-value calculations is that it is very time-consuming. Are the random permutations needed for all new applications (data)? If so, what is the time complexity? The empirical p-value accuracy depends on the # of permutations. A high # of permutations can present a more accurate empirical p-value,

but it will suffer from long computation time. On the other hand, if we reduce # of permutations, the p-value calculation loses its accuracy.

Response: Yes, the permutation test will be needed for all new applications. The time complexity for the permutation-based p -value calculations is $O(n)$ where n is the number of permutations. We agree that there is a trade-off between accuracy and computation speed for permutation-based p -value calculations. Nevertheless, as illustrated in the new supplementary Figure 3, we find a good consistency between p -values calculated by 100 and 1,000 permutations – both the slope and R-squared for the linear regression line are very close to 1 (slope=0.9987, R-squared=0.9992 for VISp projection networks; slope=0.9994, R-squared=0.9994 for ALM projection networks) (new Supplementary Figures 3a and 3b). Furthermore, the links detected by 1,000 permutations largely overlap with those detected by 100 permutations (344/346 for VISp and 359/364 for ALM), with no more than 1.5% of links missed (new Supplementary Figures 3c and 3d). These results suggest that we can practically reduce the number of permutations (e.g., 100) to save computation time while maintaining good consistency.

In the revision, these results have been added on Page 7. Related discussions are added on Pages 21-22 as follows: “There is a trade-off between accuracy and computation speed for permutation-based p -value calculations, which are needed for all new applications of NeuronChat. A high number of permutations can produce a more accurate empirical p -value, but it suffers from long computation time (the time complexity for the permutation-based p -value calculations is $O(n)$ where n is the number of permutations). Nevertheless, we find a good consistency between p -values calculated by 100 and 1,000 permutations, and the links detected by 1,000 permutations largely overlap with those detected by 100 permutations with no more than 1.5% of links missed (Supplementary Figure 3). These results suggest that one may reduce the number of permutations (e.g., default number 100 in NeuronChat) to save computation time while maintaining the accuracy of p -value calculations.”

Response to Reviewer #2

(Remarks to the Author): (A MORE READABLE PDF VERSION IS ATTACHED BELOW!)

This manuscript presents NeuronChat, a novel package for predicting and visualizing neuron-type-specific communication networks from neurotaxonomically classified single-cell RNA-seq expression data. The concept here, and as pioneered by previous studies (e.g., cited Ref. #36), is to use products of mRNA abundance metrics from ligand-receptor pairs as predictions cell-cell signaling between pairs of transcriptomically defined cell types. The method is introduced in the present work making use of a mouse cortex neurotaxonomy and genome-wide scRNA-seq datasets from cited Ref. #3 and then demonstrated by applications to human control/autism neurotaxonomies from scRNA-seq datasets from Ref. #22. Given the very large numbers of cell types and ligand-receptor pairs recognized today in virtually all metazoan brains, improved abilities to develop specific and testable hypotheses regarding communications between specific cell types seem certain to advance neuroscience.

....

In summary, this work represents an advance in a comparatively new area of quantifying interaction networks from single cell expression data. While we remain enthusiastic, the manuscript should be improved as indicated to make NeuronChat more fully useable and indicate its contribution to an exciting research approach to cell network communication

Response: We thank the reviewer for the positive evaluation and the valuable comments. Below we provide detailed responses to each specific comment.

1. A major strength of the present work is thus development of a computational package for visualizing and exploring such predictions. Unfortunately, the presentation as it stands lacks clarity at some critical junctures and leaves room for doubt as to the strength and accuracy of the "inferences" claimed as results from NeuronChat. The lack of presentation clarity is most severe in the critical section of Results entitled "Benchmarking of NeuronChat". Noting the many "false positives" evident in Figs. 2b and 2g, juxtaposed with the high incidence of "true positives" in these graphs, it is hard to see support for the statement that "NeuronChat is able to predict neural connectivity accurately and robustly".

(...particularly the part about "accurately"...). This is particularly so given that products of numerous glutamate receptor genes are expressed in virtually all CNS neurons and that Cck and Cckbr transcripts are abundant in almost all cortical glutamatergic neurons. We may be missing something here, but we fear that many other readers may have a problem with this critical passage. This section could use a very substantial re-write aimed either at more clearly leading the reader to the stated conclusion and/or softening the conclusion of "accuracy".

Response: Sorry for the confusion. In the revision, we have removed the entire sentence about "...accurately and robustly" as well as the word "accurate" or "accurately" in other parts of the manuscript. In addition, we pointed out the caveats of our predictions at several places. For example, in the section "Benchmarking of NeuronChat", we've added "Please note that a small portion of the communication links predicted are incorrect for both of the two cases (e.g., 3/21 for Figure 2b and 7/30 for Figure 2g)". Please see more updates in the section "Benchmarking of NeuronChat".

2. Another major issue is that the precise workflow is not clearly illustrated, and Figure 1 presents this workflow only in very general terms. While Figure 1 is attractive, a more detailed illustration of how interactions are represented, how gene expression is used to infer connection strength, and at least an indication of the computational model approach would help. It is difficult to understand this workflow, except at a very conceptual level, through initial text in the manuscript. For example, in the annotation of these interactions, is this a strictly binary interaction or does the data base indicate interaction strength in any way? This additional clarification to the manuscript is important as that while considerable methods are developed for the statistical methodology, the overall workflow must be first rigorously presented. The quantification model of the gene expression interactions is novel in many ways and a strength of the paper, and the permutation testing framework is an important addition to this approach.

Response: Thank you for the good suggestion. In the revision, we have modified Figure 1 to clearly illustrate the computational workflow of NeuronChat (revised Figure 1).

Specifically, in the revised Figure 1b, we show how the ligand and target abundance is estimated from gene expressions, how the permutation test is performed, and what the output communication strength matrix looks like. We have also added an example of the interaction list in Figure 1a to illustrate how ligand-target pairs in the database are curated. Correspondingly, the paragraph related to Figure 1 is modified to reflect these changes.

3. The analysis and comparison of the VISp and ALM datasets from the Allen Institute is a strength of the manuscript and offers some interesting communication links for follow up. The analysis in this part of the manuscript is generally rigorously accomplished, however, methods used for aggregating intercellular communication should be prioritized as to which are the best approaches the methods do not make this clear.

Response: We thank the reviewer for the good suggestion. To justify the choice of aggregation method for communication networks over ligand-target pairs, in the revision we have compared four different aggregation methods on inferring neuronal connectivity (new Supplementary Figure 9).

Given the sending and receiving cell group, aggregation method #1 sums the communication strength values over interaction pairs, denoted as “weight”; aggregation method #2 counts the number of links with non-zero communication strength over interaction pairs, denoted as “count”; aggregation method #3 counts the number of links with non-zero communication strength while assigning the weight of the interaction pair as the information flow, denoted as “weighted count”; aggregation method #4, denoted as “thresholded weight”, sums the communication strength values over all interaction pairs with the communication strength values filtered by a threshold for each interaction pair. For the “thresholded weight” method, we choose the threshold for an interaction pair as the 80% quantile of all communication strength values for the interaction pair. This is because the 80% quantile leads to overall higher AUROC/AUPRC than other thresholding quantiles, except a slightly lower AUROC for ALM projection network (new Supplementary Figures 9a-9b and 9e-9f).

Among the four aggregation methods, “thresholded weight” with 80% quantile produces the highest AUROC/AUPRC values for the VISp projection network (new Supplementary Figures 9c-9d), and produces the second highest AUROC/AUPRC that is only slightly lower than the best ones for the ALM projection network (new Supplementary Figures 9g-9h). Due to finite sampling in the permutation test, the predicted communication networks may fluctuate among different repeated simulations; however, “thresholded weight” leads to smaller variations in AUROC/AUPRC than other aggregation methods for repeated simulations, thus robustly minimizing the randomness generated in the permutation test (Supplementary Figure 9). Based on these observations, we choose the “thresholded weight” as the best aggregation method for benchmarking, and the other three aggregation methods are optional in our NeuronChat package.

These results have been added on Pages 11-12. Descriptions of the four methods are added to the Methods section on Page 29.

4. It also seems likely that caveats about potential disconnects between transcript metrics and abundance of proteins or their enzymatic products.

Response: Thank you for pointing out the limitation. We agree that there exist gaps between transcript levels and the abundance of proteins or metabolites. In the revision, we have added a paragraph on Page 22 to discuss the limitation and practicality of modeling protein/metabolite abundance from transcriptomics, as follows: “Like other existing methods for inferring cell-cell communications, NeuronChat estimates the abundance of ligands and target proteins from transcriptomics that could be inconsistent with protein or metabolite levels. In principle, NeuronChat can be applied to proteomics and metabolomics data to infer ligand-target interactions if the data becomes available. With the single-cell proteomics and metabolomics techniques lagging behind transcriptomics in coverage of molecules or throughput^{49, 50}, for now the transcriptomics data remain as a main data source for cell-cell communication inference.”

5. Additionally, the molecular interactions database is quite small at 373 entries and the results of the present work may be biased by this size restriction.

Response: Thanks for pointing this out. In the revision, we have added a discussion about NeuronChatDB's limited coverage and its potential effects, and we also pointed out that users can easily update the NeuronChatDB with new interactions, to expand the analysis reach. On Pages 22-23 we have added: "While NeuronChatDB includes major small-molecular neurotransmitters, most of the neuropeptides, some gasotransmitters, gap junction proteins as well as synaptic adhesion molecules, there may be missing information in the curated interaction entries, leading to bias in the inference. Nevertheless, NeuronChat allows easy updating of the database with user-defined interactions that are not included in the current version, to expand its applicability for more interactions."

6. Though the NeuronChat software package potentially offers considerable value, the present documentation falls short of enabling its effective use. The software package itself does not completely meet the standards proposed in the manuscript. There are a large number of installation dependencies in the code and these reviewers were unable to fully resolve them with the present R version. It would be most helpful to have a short example tutorial in illustrating one or more of the results of the manuscript, beyond the code the figures. If the clarity of presentation and software documentation can be improved as we suggest below and the claims of accuracy can be more clearly supported or else attenuated appropriately, NeuronChat has the potential to evolve into a valuable increment in the cellular neuroscientist's toolkit.

Response: Sorry for the confusion about the software documentation and thank the reviewer for the good suggestion. In the revised Github repository of NeuronChat (<https://github.com/Wei-BioMath/NeuronChat>), we have added a short tutorial at a conspicuous position of the README file so that users can easily view it and quickly start the NeuronChat analysis. We have also provided instructions and links to guide users for accessing a full and detailed tutorial, to reproduce results in the manuscript. To help

users for easy installation, in the revised README file, we have included detailed instructions for resolving some common issues that users may encounter during installation.

7. Finally, while we are enthusiastic about the package and approach the value of NeuronChat might be further enhanced and clarified by citing some precedents within it neuroscientific scope. One of these is already cited as Ref. #36, but the current citation only references one table that pairs neuropeptide precursors and their receptors. The authors should note that this publication is specifically about network predictions from scRNA-seq data made according to the same general logic that is at the heart of NeuronChat and directly based on the same Allen Institute neurotaxonomy and dataset as much of work presented in the current manuscript. Citation of two additional previous publications would be appropriate

1. Smith SJ, Hawrylycz M, Rossier J, Sumbul U: New light on cortical neuropeptides and synaptic network plasticity. Curr Opin Neurobiol 2020, 63:176-188. PMID: 32679509

2. Smith SJ: Transcriptomic evidence for dense peptidergic networks within forebrains of four widely divergent tetrapods. Curr Opin Neurobiol 2021, 71:100-109. PMID: 34775262

These two publications both make neurotaxonomic/transcriptomic network predictions based on the same ligand-receptor products as the present manuscript, but also point toward the necessity of prediction testing and relevant experimental and phylogenomic tests.

Response: Thank you for pointing out the three publications related to neural-specific network predictions. In the revision, we introduced the progress that has been made by the three publications on Page 3 as follows: “Smith et al. predicted 37 neuropeptide networks among cortical neuron types by taking the interaction score as the product of transcript levels of neuropeptide precursor and the cognate G-protein-coupled receptor¹⁶⁻¹⁸, but didn’t include neurotransmitter signaling.”

Response to Reviewer #3

In the manuscript “Inferring neuron-neuron communications from single-cell transcriptomics through NeuronChat”, Zhao et al. adopted a conventional cell-cell communication framework to make a specialized methodology for inferring neuron-neuron communication based on scRNA-seq data. More precisely, the method is built upon the tool CellChat which was developed and published by the same lab a while back. The authors named their new tool NeuronChat. From an application standpoint, like CellChat, which is one of the most popular tools for cell-cell communication analysis, NeuronChat is a well-documented R package, with clear tutorials accompanied. Both CellChat and NeuronChat generate very nice visualization. I do think NeuronChat is a great piece of software extension.

Response: We thank the reviewer for the positive evaluation and the valuable comments. Below we provide detailed responses to each specific comment.

1. However, almost all analysis in NeuronChat, such as latent patterns, functional similarity, have already been highlighted in the original CellChat paper. The methodological details explained in the methods section look very similar to CellChat. I feel that from a method development standpoint, with respect to CellChat, the novelty of NeuronChat is rather incremental.

Response: Thank you for pointing this out. In the revision, we have added new functionality in two ways: using spatial data for inference of cell-cell communication networks and a new multilayered visualization of spatial cell-cell communication (new Figure 7), in an added new section “NeuronChat utilizes spatial transcriptomics to infer and visualize neural-specific communication networks”.

Specifically, we have shown the added functionality using three spatial transcriptomics datasets based on three different sequencing techniques including seqFISH+, MERFISH, and Visium. The seqFISH+ dataset includes mRNA expressions of 10,000 genes in 913

cells in the mouse somatosensory cortex and subventricular zone, where there are 358 excitatory neurons of four types. The MERIFH dataset includes mRNA expressions of 258 genes in approximately 300,000 cells (including nine glutamatergic subclasses and five GABAergic subclasses as well as non-neuronal subclasses) in the mouse primary motor cortex and its adjacent areas. The Visium dataset includes mRNA expression profiles in 2,702 spots of a coronal slice of the mouse brain, and these spots are classified into seven clusters. For all three ST datasets, we first computed the communication networks among cell types (or spot clusters) without imposing spatial constraints. The neural-specific signals can transmit over long spatial distances through various physical connections among neurons that may locate far apart. To better visualize spatial communication, we have developed a new multilayered visualization functionality to illustrate together the spatial communications network, cell type/spot cluster annotation, and tissue image/ anatomic reference (new Figures 7a-7c).

To study the potential spatial effect on communication networks, next we have characterized the spatial proximity among cell types by calculating spatial proximity enrichment score similar to a previous study (Giotto). Using this information, we can remove communication links with their spatial proximity scores lower than a given threshold. Since GABAergic neurons generally have localized axonal arbors and the connection probability among them decreases with inter-neuronal distance, we showcase this functionality by applying it to communications among five GABAergic subclasses (i.e., *Lamp5*, *Sncg*, *Vip*, *Sst*, and *Pvalb*) of the MOp cortex (new Figures 7d and 7e). This spatial constraint functionality has been implemented as a user option in NeuronChat.

The above results have been added on Pages 19-20.

2. The authors emphasized that neural communication is mediated by neurotransmitters, which are non-peptides and thus excluded from most existing ligand-receptor databases. The authors therefore curated a list of neuron-specific ligand-receptors, which are not included in CellChat. Their efforts are appreciated. Nevertheless, the abundance of small molecules cannot be directly measured in scRNA-seq data, authors therefore used the

expression of the related enzymes as a proxy. While the proxy makes sense, they are also quite the obvious choice. Stoichiometric effects of metabolites and pathways dependency are not considered. There are a few recently published methods that leverage single-cell RNA-seq data to estimate metabolic flux, for instance, the tool scFEA (<https://genome.cshlp.org/content/31/10/1867>), and therefore go beyond the simple proxy used by NeuronChat.

Response: Thank you for the nice suggestion, and we agree with the reviewer's point. In the revision, we have provided a comparison between the ligand abundance by NeuronChat and scFEA-derived metabolite surrogates in identifying neural-specific communication networks (new Supplementary Figure 7). We have added related results and discussions on Pages 10-11 and Page 21, respectively.

Specifically, to investigate the effects of different metabolite surrogates in identifying neural-specific communication networks, in the revision, using glutamate as an example, we have provided a comparison between NeuronChat's ligand abundance and eight scFEA-derived surrogates (including metabolite balance and seven module fluxes) in predicting VISp and ALM projection networks. For each of the nine glutamate surrogates, we calculated AUROC and AUPRC values for the communication networks of 24 glutamate-mediated interaction pairs, and found that NeuronChat's ligand abundance shows middle or above ranking in AUROC (or AUPRC) median among the nine glutamate surrogates (new Supplementary Figures 7a and 7c). For the communication network aggregated over 24 glutamate-mediated interaction pairs, NeuronChat's ligand abundance ranks #2 in both AUROC and AUPRC for predicting VISp and ALM projecting networks among the nine glutamate surrogates (new Supplementary Figures 7b and 7d). While some of the scFEA-derived surrogates indeed show higher AUROC (or AUPRC) values than NeuronChat's ligand abundance, the difference between NeuronChat's ligand abundance and the best scFEA-derived surrogate is minimal. These results indicate that NeuronChat's ligand abundance works relatively well despite its simplicity.

The results have been added on Pages 10-11. Related discussions have been added on Page 22 as follows: “While NeuronChat’s computational workflow has been optimized to predict neuronal connectivity, the settings can be expanded to incorporate more refined models, for example, for estimating the abundance of small molecular neurotransmitters. For such cases, the stoichiometric effects of metabolism and pathways dependency may be included in addition to using expressions of only synthetic enzymes and vesicular transporters. By comparing nine glutamate abundance surrogates, we find that some of the scFEA-derived surrogates show higher AUROC (or AUPRC) values than NeuronChat’s ligand abundance (Supplementary Figure 7). While the difference between NeuronChat’s ligand abundance and the best scFEA-derived surrogate is small, it suggests ways in improving the prediction accuracy of neuronal connectivity.”

3. As the only quantitative justification of NeuronChat, the authors used two projective networks identified using retrograde labeling as the gold standard. Even though a reference is provided, it seems that the details of the two networks were not mentioned. I assume retrograde labeling measure connections in a single-cell level, but NeuronChat predicts connections in cell-type level. It is not clear how sensitivity and specificity are defined. As the topology of the gold standard networks might affect the results (for instance, a cell-type is connected to many other cell-type, making prediction easier), the current AUC values (0.83, 0.76 etc.) will make more sense if the authors could repeat the analysis but shuffling the edges in the gold standard networks. I wonder if the resultant AUCs would reduce to 0.5. If not, I am not sure how to interpret the current AUC values, which are already not too impressive.

Response: We thank the reviewer for the good suggestion of additional benchmarking. To determine whether the specific graph topology of ground truth labels makes the prediction task easy for NeuronChat, in the revision we perturb the ground truth labels by shuffling cell type labels of graph nodes while keeping the same graph topology, and then calculate AUROC and AUPRC (new Supplementary Figure 1). We find that the AUROC for the shuffled ground truth labels leads to a distribution with a mean around 0.5, indicating a poor prediction ability for those shuffled labels with even the same topology

(new Supplementary Figures 1a, 1c, 1e, and 1g). We also show that the AUROC for the original ground truth labels is significantly higher than those for shuffled labels (p -values are 0.010 ± 0.0036 and 0.017 ± 0.0048 for VISp and ALM projection networks, respectively). Similar results are also obtained for the calculation of AUPRC (new Supplementary Figures 1b, 1d, 1f, and 1h), suggesting NeuronChat's prediction ability doesn't directly reflect the specific graph topology of ground truth labels. These new results have been added on Page 6.

To address the reviewer's concerns on the two projection networks, in the revision we have added detailed descriptions regarding how retrograde labeling identifies neural connections and how the coarse-grained projection networks at the cell-type level are obtained. Please see the description on Pages 5-6: "The connections from excitatory neurons of VISp and ALM to their cortical target regions were identified using monosynaptic retrograde labeling³, where the viral tracers are injected into target regions and move towards the presynaptic neurons via retrograde axonal transport without further spreading to indirectly contacted cells, allowing the identification of direct neural connections²¹⁻²⁵. By grouping retrogradely labeled neurons using their cell type annotations, we obtain the coarse-grained projection networks composed of directed links from excitatory neuron types in VISp and ALM to their cortical target regions (Figure 2a for VISp and Figure 2f for ALM), which are then used for subsequent benchmarking."

Revised Figure 1. Overview of NeuronChat.

(a) Overview of NeuronChat database. NeuronChat database includes ligand-target pairs required for chemical synapse, electrical synapse and synaptic adhesion (left panel). There are a total of 373 ligand-target pairs for both human and mouse, curated into five categories based on the type of the ligand (middle panel). The interaction pair list includes the ligand, target, and genes contributing to them (right panel). Note that genes contributing to the ligand are categorized into different groups (indicated by colors) based on their biological functions such as synthesis or vesicular transport.

(b) Schematic diagram to illustrate the computational model of NeuronChat. The communication strength characterizes the coordinated expression of genes required for ligand emission in the sender cell group, and the expression of the target gene in the receiver cell group. The statistical significance of a communication link is determined by the permutation test (* and ns represent significant and not significant, respectively). Only significant links are kept in the output communication strength matrix while values for not significant links are set to be zeros. See Methods for details.

(c) Functionalities of NeuronChat: visualization and analysis of the intercellular communication networks, making systemic comparisons across different biological contexts, and multi-layered visualization for spatial transcriptomics.

New Figure 3. Comparison of NeuronChat, CellPhoneDB, and CellChat in predicting VISp and ALM projection networks.

(a) Typical ROC curves (left panel) and PR curves for the three methods.

(b) The boxplots of AUROC (left panel) and AUPRC (right panel) values for 100 repeats of the aggregated VISp projection networks inferred by the three methods. Each boxplot represents 100 independent repeated computations. Boxplot elements: center line, median; box limits, upper and lower quartiles; whiskers, 1.5x interquartile range; points, outliers. Note that no variation in each boxplot is observed because the aggregated method ‘thresholded weight’ reduces the fluctuation caused by finite sampling in the permutation test (see also Supplementary Figure 9).

(c) The number of detected interaction pairs for the three methods.

(d-e) The boxplots of AUROC (d) and AUPRC (e) values for the individual VISp projection networks inferred by the three methods. Boxplot elements: center line, median; box limits, upper and lower quartiles; whiskers, 1.5x interquartile range; points, outliers.

(f-j) Repeat analysis for ALM projection network, analogous to (a-e).

New Figure 7. Multi-layered visualization for spatial data and inference of spatially constrained communication network

(a-c) Multi-layered visualization for three spatial transcriptomics datasets generated by different techniques seqFISH+ (a), MERFISH (b), and Visium (c). Each plot includes the raw tissue slice image/anatomic reference (bottom), cell/spot annotation in space (middle), and the aggregated communication network with the top 10 links shown (top). The width of a link indicates the sum of communication strengths over all significant ligand-target pairs. See Supplementary Figure 12 for the full aggregated networks. The bottom image in (b) is the brain anatomic reference (Image credit: Allen Institute for Brain Science. [<http://atlas.brain-map.org/atlas?atlas=1#atlas=1&plate=100960348>]).

(d) Bar plot showing the cell proximity enrichment scores for all pairwise interacting cell types. The cell proximity enrichment scores are calculated based on all 64 MERFISH slices. The score > 0 (bars in red) and score < 0 (bars in cyan) represent enriched and depleted proximity between interacting cell types, respectively.

(e) The inference of spatially constrained communication network for GABAergic neurons. Left panel: the cell proximity network. Links in red or grey represent enriched or depleted proximity between interacting cell types, respectively; the width of a link indicates the strength of enrichment or depletion. Middle panel: cell-cell communication network without spatial constraint, calculated based on scRNA-seq data²⁶ for the same brain region and same cell types (2,044 single cells in total). The width of a link indicates the sum of communication strengths over all significant ligand-target pairs. Right panel: the spatially constrained cell-cell communication network, obtained by removing links with depleted proximity from the original cell-cell communication network. See also Supplementary Figure 13.

New Supplementary Figure 1. Benchmarking NeuronChat with shuffled ground truth labels.

(a-b) The distribution of AUROC (a) and AUPRC (b) values for 1,000 times of cell type label shuffling in the VISp projection network. The red line indicates the original AUROC/AUPRC for the VISp projection network without label shuffling.

(c-d) The distribution of p -values for AUROC (c) and AUPRC (d). The p -value is defined as the proportion of AUROC/AUPRC values that are larger than or equal to the original one (indicated by the red line in a-b), based on 100 independent repeated simulations. Mean \pm SD for the p -values is 0.010 \pm 0.0036 for AUROC and 0.012 \pm 0.0038 for AUPRC.

(e-h) Repeat analysis for ALM projection network, analogous to (a-d). Mean \pm SD for the p -values is 0.017 \pm 0.0048 for AUROC and 0.0087 \pm 0.0033 for AUPRC.

New Supplementary Figure 3. Comparison of p -values and number of links calculated by 100 and 1,000 permutations.

(a-b) The scatter plot of p -values calculated by 100 and 1,000 permutations for VISp (a) and ALM (b) projection network. The dots in each plot represents all non-zero cell-cell communication links for all possible interaction pairs. The plot shows the permutation test's original p -values that are not adjusted by Benjamini-Hochberg procedure. The regression line, regression equation, and adjusted R-squared are shown in each plot.

(c-d) Comparison of the number of links detected by 100 and 1,000 permutations for VISp (c) and ALM (d) projection network.

New Supplementary Figure 7. Comparison between NeuronChat’s ligand abundance and scFEA-derived metabolite surrogates in identifying glutamate-mediated communication networks.

(a) Boxplots of AUROC (upper panel) and AUPRC (lower panel) values of 24 glutamate-mediated communication networks for predicting the VISp projection network, for the nine glutamate surrogates. For the metabolic module representing the opposite direction of glutamate accumulation (e.g., M_48), we use the maximum flux value among all cells minus the original flux as the surrogate for glutamate, denoted by the original module name with a suffix “r” (same for b, c, and d). Boxplot elements: center line, median; box limits, upper and lower quartiles; whiskers, 1.5x interquartile range; points, outliers.

(b) Barplots of AUROC (upper panel) and AUPRC (lower panel) values of the aggregated glutamate-mediated communication networks for predicting VISp projection network, for the nine glutamate surrogates.

(c-d) Repeat analysis for ALM projection network, analogous to (a-b).

New Supplementary Figure 8. Comparison of Tukey's trimean and arithmetic mean in predicting VISp and ALM projection networks.

- (a) The number of detected interaction pairs for the VISp projection networks for the two mean methods.
- (b-c) Boxplots showing the AUROC (b) and AUPRC (c) values of the individual VISp projection networks for the two mean methods. Boxplot elements: center line, median; box limits, upper and lower quartiles; whiskers, 1.5x interquartile range; points, outliers.
- (d) Distributions of AUROC (left panel) and AUPRC (right panel) values of the individual VISp projection networks for the two mean methods.
- (e-h) Repeat analysis for ALM projection network, analogous to (a-d).

New Supplementary Figure 9. Optimization of the aggregation method.

(a-b) Boxplots of AUROC (a) and AUPRC (b) values for 100 repeats of the aggregated VISp projection networks at different quantile thresholds of the aggregation method “thresholded weight”. Boxplot elements: center line, median; box limits, upper and lower quartiles; whiskers, 1.5x interquartile range; points, outliers.

(c-d) Boxplots of AUROC (c) and AUPRC (d) values for 100 repeats of the aggregated VISp projection networks for four different aggregation methods. For the aggregation method “thresholded weight”, the thresholding parameter is chosen as 80% quantile of communication strength values for each interaction pair. Boxplot elements: center line, median; box limits, upper and lower quartiles; whiskers, 1.5x interquartile range; points, outliers.

(e-h) Repeat analysis for ALM projection network, analogous to (a-d).

New supplementary Figure 12. Inference of communication network for three spatial transcriptomics datasets.

(a-c) The raw tissue slice image/brain region annotation diagram (first panel), spatial map (second panel), full aggregated communication network (third panel), and aggregated communication network with top 10 links shown (fourth panel), for three spatial transcriptomics datasets generated by different techniques including seqFISH+ (a), MERFISH (b), and Visium (c). In spatial maps, a dot represents the centroid of a cell (for a-b) or a spot (for c). The communication network summarizes the communication strength over interaction pairs (i.e., the “weight” aggregation method). For seqFISH+ and Visium datasets, the communication networks are directly calculated from the spatial transcriptomics; for the MERFISH dataset, the communication network is calculated using single-cell RNA-seq data (4,461 cells of seven glutamatergic subclasses) for the same brain region. For (b), the brain region annotation image (Image credit: Allen Institute for Brain Science. <http://atlas.brain-map.org/atlas?atlas=1#atlas=1&plate=100960348>) highlights the mouse primary motor cortex region (in purple); the spatial map only shows the glutamatergic cells in one representative coronal slice (slice id: mouse1_slice212).

New supplementary Figure 13. The spatial map of GABAergic neurons for the MERFISH dataset.

- (a) Spatial map of five subclasses of GABAergic neurons in a coronal slice (slice id: mouse1_slice212). A dot represents the centroid of a cell.
- (b) Distribution of the y-axis coordinate for the five GABAergic subclasses shown in (a). The y-axis coordinate roughly represents the range from cortical layer L1 (top) to L6b (bottom).

REVIEWERS' COMMENTS

Reviewer #1 (Remarks to the Author):

The authors have addressed most of my major comments. Specifically, 1) the authors added new analyses on single-cells spatial data. A spatial constraint functionality was also added as a function of neuronchat. 2)The benchmarking analyses were also added, as suggested. The neuronchat was compared to state-of-the-art tools such as Cellphonedb and Cellchat, and demonstrated superior performance. All other minor comments were also addressed. I have no further comments and now support the publication of this manuscript.

Reviewer #2 (Remarks to the Author):

The authors are to be congratulated on a very responsive revision of a manuscript which now very clearly describes the development and use of an excellent software tool likely to be of great use in neuroscience.

Reviewer #3 (Remarks to the Author):

The authors have substantially improved the manuscript with new spatial omics data applications. All my concerns were well addressed. It will be a useful tool for neuroscience field and beyond.

Response to Reviewers #1, #2 and #3

REVIEWERS' COMMENTS

Reviewer #1 (Remarks to the Author):

The authors have addressed most of my major comments. Specifically, 1) the authors added new analyses on single-cells spatial data. A spatial constraint functionality was also added as a function of NeuronChat. 2)The benchmarking analyses were also added, as suggested. The NeuronChat was compared to state-of-the-art tools such as CellPhoneDB and CellChat, and demonstrated superior performance. All other minor comments were also addressed. I have no further comments and now support the publication of this manuscript.

Reviewer #2 (Remarks to the Author):

The authors are to be congratulated on a very responsive revision of a manuscript which now very clearly describes the development and use of an excellent software tool likely to be of great use in neuroscience.

Reviewer #3 (Remarks to the Author):

The authors have substantially improved the manuscript with new spatial omics data applications. All my concerns were well addressed. It will be a useful tool for neuroscience field and beyond.

Response: We thank the reviewers for the positive evaluation and the helpful suggestions in the first round of review. Since all reviewers have no further comments, in this revision we only revised the manuscript to comply with the editorial requests.